# On Equivalences between Weight and Function-Space Langevin Dynamics

## Abstract

Approximate inference for overparameterized Bayesian models appears challenging, due to the complex structure of the posterior. To address this issue, a recent line of work has investigated the possibility of directly conducting approximate inference in the "function space", the space of prediction functions. This paper provides an alternative perspective to this problem, by showing that for many models – including a simplified neural network model – Langevin dynamics in the overparameterized "weight space" induces equivalent function-space trajectories to certain Langevin dynamics procedures in function space. Thus, the former can already be viewed as a function-space inference algorithm, with its convergence unaffected by overparameterization. We provide simulations on Bayesian neural network models and discuss the implication of the results.

## 1 Introduction

Consider a common Bayesian predictive modeling setting: we are provided with i.i.d. observations $\mathcal{D} := \{(x_i, y_i)\}_{i=1}^n$ where $x_i \in \mathcal{X}$, $y_i \in \mathbb{R}$ and $\mathcal{X}$ denotes the input space; a likelihood model $p(\{y_i\} \mid \{x_i\}, \theta) = \prod_{i=1}^n p(y_i \mid f(x_i; \theta))$ determined by a prediction function $f(\cdot; \theta)$; and a prior $\pi_\theta(\mathrm{d}\theta)$. We are interested in the predictive distribution $p(y_* \mid x_*, \mathcal{D}) = \int \pi_{\theta|\mathcal{D}}(\mathrm{d}\theta) p(y_* \mid x_*, \theta)$, induced by the posterior $\pi_{\theta|\mathcal{D}}$.

Modern machine learning models are often overparameterized, meaning that multiple parameters may define the same likelihood. For example, in Bayesian neural network (BNN) models where $\theta \in \mathbb{R}^d$ denote the network *weights*, we can obtain a combinatorial number of equivalent parameters by reordering the neurons, after which $f(\cdot; \theta)$, and thus the likelihood, remain unchanged. Consequently, the posterior measure exhibits complex structures and becomes hard to approximate; for example, its Lebesgue density may contain a large number of global maxima.

Starting from Sun et al. (2019); Wang et al. (2019); Ma et al. (2019), a recent literature investigates the possibility of simplifying inference by approximating a *function-space posterior*. Concretely, let $\mathcal{A} : \mathbb{R}^d \to \mathcal{F} \subset \mathbb{R}^\mathcal{X}, \theta \mapsto f(\cdot; \theta)$ denote a "parameterization map". Then

$$p(y_* \mid x_*, \mathcal{D}) = \int \pi_{\theta|\mathcal{D}}(\mathrm{d}\theta) \, p(y_* \mid f(x_*; \theta)) = \int (\mathcal{A}_\# \pi_{\theta|\mathcal{D}})(\mathrm{d}f) \, p(y_* \mid f(x_*)) = \int \pi_{f|\mathcal{D}}(\mathrm{d}f) \, p(y_* \mid f(x_*)),$$

where $\mathcal{A}_\#(\cdot)$ refers to the pushforward measure (Villani, 2009, p. 11), and $\pi_{f|\mathcal{D}}$ denotes the function-space posterior defined by the prior $\mathcal{A}_\# \pi_\theta =: \pi_f$ and likelihood $p(y \mid x, f) = p(y \mid f(x))$. As shown above, $\pi_{f|\mathcal{D}}$ is sufficient for prediction. Moreover, it often has simpler structures: for example, for ultrawide BNN models with a Gaussian $\pi_\theta$, $\pi_f$ may converge to a Gaussian process (GP) prior (Lee et al., 2018; Matthews et al., 2018; Yang, 2019), in which case $\pi_{f|\mathcal{D}}$ will also converge to a GP posterior. Thus, it is natural to expect approximate inference to be easier in function space.

While the intuition has been appealing, existing works on function-space inference tend to be limited by theoretical issues: principled applications may require full-batch training (Sun et al., 2019), Gaussian likelihood (Shi et al., 2019), or specifically constructed models (Ma et al., 2019; Ma & Hernández-Lobato, 2021). Many approaches rely on approximations to the function-space prior, which can make the functional KL divergence

unbounded (Burt et al., 2020). Additionally, there is a lack of understanding about optimization convergence, or the expressivity of the variational families used. In contrast, gradient-based MCMC methods, such as Hamiltonian Monte Carlo (HMC) or Langevin dynamics (LD)-based algorithms, can be applied to a broad range of models. Their convergence behaviors are well-understood (Roberts & Tweedie, 1996; Villani, 2009), and intriguingly, their performance often appears to be satisfying on massively overparameterized models (Zhang et al., 2019; Izmailov et al., 2021), even though they are implemented in weight space.

This paper bridges the two lines of approaches by showing that

- In various overparameterized models, including two simplified BNN models (Sec. 2.1 and Ex. 2.3), weight-space Langevin dynamics (LD) is equivalent to a reflected / Riemannian LD procedure in function space, defined by the pushforward metric.

- For practical feed-forward network models, *a possible consequence* of the equivalence still appears to hold in simulations (Sec. 3): weight-space LD produces predictive distributions that appears to approach the functional posterior, at a rate that does not depend on the degree of overparameterization.

The equivalence has important implications: it means that *principled function-space inference has always been possible and in use.* Thus, explicit consideration of function-space posteriors *alone* will not be sufficient to guarantee improvement over existing approaches, and more careful analyses are necessary to justify possible improvement. We also discuss how further insights into the behavior of weight-space LD could be gained by comparing the pushforward metric with the prior (Sec. 2.2).

It should be noted that in several scenarios, it has been established that overparameterization does not necessarily hinder the convergence of LD. Moitra & Risteski (2020) proves that polynomial convergence can be possible for a family of *locally* overparameterized models, despite the non-convexity introduced by the over-parameterization.[1] Dimensionality-independent convergence has also been established for infinite-width NNs in the mean-field regime (e.g., Mei et al., 2019), even though its implication for practical, finite-width models is less clear. More broadly, at a high level our work is also related to past works that studied alternative inference schemes for different models that exhibit some redundancy in the parameterization (Papaspiliopoulos et al., 2007; Yu & Meng, 2011). We are unaware of strict equivalence results as provided in this paper, but we should also emphasize that it is not their technical sophistication that makes them interesting; it is rather *their implications for BNN inference, which appear underappreciated*: the results justify the use of LD as an effective function space inference procedure, in settings that match or generalize previous work. For example, Example 2.1 covers overparameterized linear models, and many popular approaches (e.g., Osband et al., 2018; He et al., 2020) are only justified in this setting.

Our results contribute to the understanding of the real-world performance of BNN models, as they provide a theoretical support for the hypothesis that inference may be good enough in many applications, and is not necessarily the limiting factor in a predictive modeling workflow. In this aspect, our results complement a long line of existing work which examined the influence of likelihood, prior and data augmentation in BNN applications, with an emphasis on classification tasks with clean labels; see Aitchison (2020); Wenzel et al. (2020); Fortuin et al. (2021), to name a few.

## 2 Equivalence between Weight and Function-Space Langevin Dynamics

Suppose the prior measure $\pi_\theta$ is supported on an open subset of $\mathbb{R}^d$ and has Lebesgue density $p_\theta$. The weight-space posterior $\pi_{\theta|\mathcal{D}}$ can be recovered as the stationary measure of the (weight-space) Langevin dynamics

$$\mathrm{d}\theta_t = \nabla_\theta(\log p(\mathbf{Y} \mid \theta_t, \mathbf{X}) + \log p_\theta(\theta_t))\mathrm{d}t + \sqrt{2}\mathrm{d}B_t, \tag{WLD}$$

where we write $\mathbf{X} := \{x_i\}_{i=1}^n, \mathbf{Y} := \{y_i\}_{i=1}^n$ for brevity.

---

[1]This result is still not fully unimpeded by overparameterization, as it quantifies convergence to the weight-space posterior, which necessarily requires traversal through all symmetric regions.

The pushforward measure $\mathcal{A}_{\#}\pi_\theta =: \pi_f$ provides a prior in function space. Combining $\pi_f$ and the likelihood leads to a posterior, $\pi_{f|\mathcal{D}}$. When the function space $\mathcal{F} := \operatorname{supp}\pi_f$ is a Riemannian manifold[2] of dimensionality $k \leq d$, it is intuitive that we could sample from $\pi_{f|\mathcal{D}}$ by simulating a Riemannian Langevin dynamics on $\mathcal{F}$ (Girolami & Calderhead, 2011). In coordinate form:

$$\mathrm{d}\tilde{f}_t = V(\tilde{f}_t)\mathrm{d}t + \sqrt{2G^{-1}(\tilde{f}_t)}\mathrm{d}B_t, \tag{FLD}$$

where $\tilde{f}_t \in \mathbb{R}^k$ is the coordinate of $f_t \in \mathcal{F}$, $G^{-1}(\tilde{f}) = (g^{ij})_{i,j\in[k]}$ is the inverse of the metric matrix and $g_{ij}$ are the local representations for the metric (Lee, 2018, p. 13), $\mathrm{d}B_t$ is the standard Brownian motion, and

$$V^i(\tilde{f}) = \sum_{j=1}^k g^{ij}\partial_j\left(\log p(\mathbf{Y} \mid \tilde{f}, \mathbf{X}) + \log\frac{d\pi_f}{d\mu_{\mathcal{F}}}(\tilde{f}) - \frac{\log|G|}{2}\right) + \sum_{j=1}^k \partial_j g^{ij}.$$

In the above, $\mu_{\mathcal{F}}$ denotes the corresponding *Riemannian measure* (Do Carmo, 1992, p. 45), and $p(\mathbf{Y} \mid \tilde{f}, \mathbf{X}) := p(\mathbf{Y} \mid f(\mathbf{X}))$ denotes the likelihood of the function $f$ corresponding to $\tilde{f}$.

We are interested in possible equivalences between the induced function-space trajectory of (WLD), $\{\mathcal{A}\theta_t\}$, and the trajectory of possibly generalized versions of (FLD), with metric defined as the pushforward of the Euclidean metric by $\mathcal{A}$ or its generalization. By equivalence we mean that for any $k \in \mathbb{N}$ and $\{t_i\}_{i=1}^k \subset [0,\infty)$, $\{\mathcal{A}\theta_{t_i}\}_{i=1}^k$ and $\{f_{t_i}\}_{i=1}^k$ equal in distribution. When it holds, an algorithm that simulates (WLD) for a time period of $T$ and returns $\mathcal{A}\theta_T$ *can be equivalently viewed as a "function-space inference algorithm"*, as it is then equivalent (in distribution) to the simulation of (FLD) which does not have to be defined w.r.t. an overparameterized model.

We will first illustrate the equivalence on linear models (Sec. 2.1) which, while technically simple, provides intuition and formally covers NN models in the "kernel regime" (Woodworth et al., 2020). We will then discuss the role of the pushforward metric in (FLD) (Sec. 2.2), and analyze general models in which overparameterization can be characterized by group actions (Sec. 2.3).

## 2.1 Overparameterized Linear Models

The following is the easiest example where the equivalence can be demonstrated:

**Example 2.1** (equivalence in linear models; see Appendix B.1 for details)**.** *Suppose the map $\mathcal{A}$ is linear. For expository simplicity, further assume that $\pi_\theta = \mathcal{N}(0, I)$, and that the input space $\mathcal{X} = \{\mathbf{x}_1, \mathbf{x}_2, \ldots, \mathbf{x}_K\}$ has finite cardinality $K$, so that any function can be represented by a $K$ dimensional vector $(f(\mathbf{x}_1), \ldots, f(\mathbf{x}_K))$, and $\mathcal{A}$ can be identified as a matrix $A \in \mathbb{R}^{K \times d}$.*

*(i) If $\mathcal{A}$ is a bijection (i.e., $d = K$ and $A$ is invertible), the above vector representation will provide a coordinate for $\mathcal{F}$. In this coordinate, the metric matrix $G$ is $(AA^\top)^{-1}$ (see e.g., Bai et al., 2022). (FLD) with this metric reduces to*

$$\mathrm{d}\tilde{f}_t = (AA^\top)\nabla_{\tilde{f}}\left(\log p(\mathbf{Y} \mid \tilde{f}, \mathbf{X}) - \frac{1}{2}\|A^{-1}\tilde{f}_t\|_2^2\right)\mathrm{d}t + \sqrt{2AA^\top}\mathrm{d}B_t. \tag{1}$$

*By Itô's lemma, the above SDE also describes the evolution of $A\theta_t$, for $\theta_t$ following (WLD).*

*(ii) The equivalence continue to hold in the overparameterized case (e.g., when $d > K$): consider the decomposition $\mathbb{R}^d = \operatorname{Ran}(A^\top)\oplus\operatorname{Ker}(A)$. Then the evolution of $\theta_t$ in (WLD) "factorizes" along the decomposition: the likelihood gradient is fully contained in $\operatorname{Ran}(A^\top)$ and thus only influences $\operatorname{Proj}_{\operatorname{Ran}(A^\top)}\theta_t$, whereas $\operatorname{Proj}_{\operatorname{Ker}(A)}\theta_t$ has no influence on $\mathcal{A}\theta_t$. Therefore, we can describe the evolution of the former independently, thereby reducing to the exactly parameterized case.*

The second case above provides the first intuition on why (WLD) is not necessarily influenced by overparameterization. While technically simple, it is relevant as it covers random feature models, which only require

---

[2]See Appendix A.2 for a conceptual review of relevant notions in Riemannian geometry.

replacing $\mathbf{X}$ with preprocessed features. Random feature models formally include infinitely wide DNNs in the "kernel regime" (Jacot et al., 2018), where the pushforward metric converges to a constant value. As referenced before, many popular procedures for BNN inference are only justified in this regime.

## 2.2 The Pushforward Metric

The pushforward metric that instantiates our (FLD) is an important object in the study of DNNs, in which it is named the "neural tangent kernel" (NTK, Jacot et al., 2018). It acts as a preconditioner in our function-space dynamics, and makes a similar appearance in the analysis of gradient descent (GD) where its preconditioning effect is often believed to be desirable (Arora et al., 2019a;b; Lee et al., 2019).

As cited before, for BNN models with a Gaussian $\pi_\theta$, the function-space prior can converge to a Gaussian process (the "NNGP", Lee et al., 2018) as the network width goes to infinity. The NTK is closely related to the covariance kernel of the NNGP; they are equivalent if only the last layer of the DNN is learnable, and for more general models may still share the same Mercer eigenfunctions (Arora et al., 2019a, App. H). *When the two kernels are close* and the BNN model is correctly specified, it can be informally understood that (FLD) *may enjoy good convergence properties*, by drawing parallels to the analyses of GD (Arora et al., 2019a; Lee et al., 2019);[3] consequently, the approximate posterior will have a good predictive performance. However, for very deep networks, the Mercer spectra of the two kernels can be very different (Arora et al., 2019a, Fig. 4), in which case we can expect (FLD) to have poor convergence.

*The above discussions immediately apply to* (WLD) when it is equivalent to (FLD) or its suitable variants. More generally, however, it can still be helpful to check for significant differences between the NNGP and NTK kernels when using (WLD), as part of a prior predictive check (Box, 1980) process. This is especially relevant for deeper models, because in certain initialization regimes, both kernels can have pathological behavior as the network depth increases (Schoenholz et al., 2016; Hayou et al., 2019).

## 2.3 Overparameterization via Group Actions

It is often the case that overparameterization can be characterized by group actions; in other words, there exists some group $H$ on $\mathbb{R}^d$ s.t. any two parameters $\theta, \theta' \in \mathbb{R}^d$ induce the same function $\mathcal{A}\theta = \mathcal{A}\theta'$ if and only if they belong to the same orbit. In such cases, we can identify $\mathcal{F}$ as the quotient space $\mathbb{R}^d/H$ and the map $\mathcal{A} : \mathbb{R}^d \to \mathcal{F}$ as the quotient map, and it is desirable to connect (WLD) to possibly generalized versions of (FLD) on $\mathcal{F}$. This subsection presents such results.

To introduce our results, we first recall some basic notions in group theory. (Additional background knowledge is presented in Appendix A.) Let $H$ be a Lie group. The unit element of $H$ is denoted as $e$, and we use $\varphi_1\varphi_2 \in H$ to denote the group operation of $\varphi_1, \varphi_2 \in H$. An *action* of $H$ on $\mathbb{R}^d$ is a map $\Gamma : H \times \mathbb{R}^d \to \mathbb{R}^d$, s.t. for all $\varphi_1, \varphi_2 \in H$ and $p \in \mathbb{R}^d$, we have $\Gamma(e, p) = p, \Gamma(\varphi_1, \Gamma(\varphi_2, p)) = \Gamma(\varphi_1\varphi_2, p)$ where $e \in H$ denotes the identity. We use $\varphi \cdot p$ to denote $\Gamma(\varphi, p)$ for simplicity. For any $\varphi \in H$, introduce the map $\Gamma_\varphi : \mathbb{R}^d \to \mathbb{R}^d, p \mapsto \varphi \cdot p$. Then the action is *free* if $\Gamma_\varphi$ has no fixed point for all $\varphi \neq e$, *proper* if the preimage of any compact set of the map $(\varphi, p) \mapsto \varphi \cdot p$ is also compact, and *smooth* if $\Gamma_\varphi$ is smooth for each $\varphi \in H$. An *orbit* is defined as $H \cdot p := \{\varphi \cdot p : \varphi \in H\}$ where $p \in \mathbb{R}^d$.

**Analysis of free group actions** The quotient manifold theorem (Lee, 2012, Theorem 21.10) guarantees that the quotient space $\mathbb{R}^d/H$ is a smooth manifold if the action is smooth, proper and free. To define the pushforward metric on $\mathcal{F}$, we further assume that the action is isometric, i.e., $\Gamma_\varphi$ is an isometry for every $\varphi \in H$. Under this condition, a metric on $\mathcal{F}$ can be defined as[4]

$$\langle (\mathrm{d}\mathcal{A}|_p)(u), (\mathrm{d}\mathcal{A}|_p)(v) \rangle_{T_{\mathcal{A}p}\mathcal{F}} := \langle u, v \rangle_{\mathbb{R}^d}, \quad \forall p \in \mathcal{F}, \ u, v \in T_p(H \cdot p)^\perp \subset \mathbb{R}^d.$$

---

[3]For the kernel regime and a Gaussian likelihood, a precise analysis can be possible: the evolution of $f_t(\mathbf{X})$ factorizes along the eigenvectors of the NTK Gram matrix. We forgo it for brevity.

[4]It is well-defined since $\mathrm{d}\mathcal{A}|_p$ is an isomorphism between $T_p(H \cdot p)^\perp$ and $T_{\mathcal{A}p}\mathcal{F}$, and the isometry assumption ensures that the definition is independent of the choice of $p$ in the orbit (Lee, 2018).

The above equation used some standard notations in differential geometry (Lee, 2018, p. 16): $\mathrm{d}\mathcal{A}|_p : \mathbb{R}^d \to T_{\mathcal{A}p}\mathcal{F}$ is the *differential* of $\mathcal{A}$ at $p$, $T_p$ denotes the *tangent space* of a manifold, and $T_p(H \cdot p)^\perp$ is the orthogonal complement of the tangent space of the orbit $H \cdot p$, which is a submanifold of $\mathbb{R}^d$.

The following proposition establishes the equivalence under discrete group action.

**Proposition 2.1** (proof in Appendix B.2)**.** *Suppose $H$ is a discrete group (Hall, 2013, p. 28) acting smoothly, freely, properly on $\mathbb{R}^d$, and $\mathcal{A}$ is such that $\mathcal{A}\theta = \mathcal{A}\theta'$ if and only if $\theta' \in H \cdot \theta$. If either (a) the (improper) prior $p_\theta$ is constant and the group action is isometric; or (b) $H = \{e\}$ is trivial, then the equivalence between* (WLD) *and* (FLD) *will hold.*

*Remark* 2.1. For continuous groups that act freely, the situation is more complicated, and depends on how the orbits are embedded in the ambient space $\mathbb{R}^d$. For example, a drift term depending on the mean curvature vector of the orbit may be introduced when pushing a Brownian motion using the quotient map (JE, 1990), and when the mean curvature vanishes, the equivalence will continue to hold, as shown in our Example 2.1 (ii). Analysis for non-free group actions is primarily complicated by the fact that the quotient space is no longer a manifold in general (Satake, 1956). Still, as we show in Example 2.3, similar results can be established under the action of symmetric groups.

We now provide a concrete, albeit artificial, example in which the equivalence implies fast convergence to the function-space posterior. It also highlights that VI and MCMC methods can have different behavior on overparameterized models, and that for VI methods it may still be necessary to explicitly account for overparameterization. While recent works have made similar observations (e.g., Sun et al., 2019), and provided some examples (Wang et al., 2019; Kurle et al., 2022), our example may provide additional insight:

**Example 2.2** (LD vs. particle-based VI on torus)**.** *Let $\mathcal{A}\theta := ([\theta_1], \ldots, [\theta_d])$, where $[a] := a - \lfloor a \rfloor \in [0, 1)$. Let $\pi_\theta, \pi_f$ have constant densities, and the negative log likelihood be unimodal and locally strongly convex. Then we have $\mathcal{F} = \mathbb{T}^d$, the $d$-dimensional torus, and by Proposition 2.1,* (WLD) *is equivalent to Riemannian LD on $\mathcal{F}$. As $\mathbb{T}^d$ is a compact manifold,* (FLD) *enjoys exponential convergence (Villani, 2009), and so does the induced function-space measure of* (WLD)*.*

*Particle-based VI methods approximate the weight-space posterior with an empirical distribution of particles $\{\theta^{(i)}\}_{i=1}^M$, and update the particles iteratively. Consider the W-SGLD method in Chen et al. (2018): its update rule resembles* (WLD)*, but with the diffusion term replaced by a deterministic "repulsive force" term, $\tilde{v}_t(\theta)\mathrm{d}t$, where*

$$\tilde{v}_t(\theta) := \sum_{j=1}^M \frac{\nabla_{\theta^{(j)}} k_h(\theta, \theta^{(j)})}{\sum_{k=1}^M k_h(\theta^{(j)}, \theta^{(k)})} + \frac{\sum_{j=1}^M \nabla_{\theta^{(j)}} k_h(\theta, \theta^{(j)})}{\sum_{k=1}^M k_h(\theta, \theta^{(k)})},$$

*and $k_h$ is a radial kernel with bandwidth $h$. Formally, in the infinite-particle, continuous time limit, as $h \to 0$, both $\tilde{v}_t\mathrm{d}t$ and the diffusion term implements the Wasserstein gradient of an entropy functional (Carrillo et al., 2019), and W-SGLD and LD are formally equivalent (Chen et al., 2018).*

*The asymptotic equivalence between* (WLD) *and W-SGLD breaks down in this example: whereas* (WLD) *induces a function-space measure that quickly converges to $\pi_{f|\mathcal{D}}$, this is not necessarily true for W-SGLD. Indeed, its induced function-space measure may well collapse to a point mass around the MAP, regardless of the number of particles. To see this, let $\theta^* \in [0, 1)^d$ be any MAP solution so that $\nabla_\theta \log p(\mathbf{Y} \mid \mathbf{X}, \theta^*)p(\theta^*) = 0$. Then for any fixed $h = O(1)$, as $M \to \infty$, the configuration $\{\theta^{(i,M)} = (10^{10Mi}, 0, \ldots, 0) + \theta^*\}_{i=1}^M$ will constitute an approximate stationary point for the W-SGLD update. This is because the posterior gradient term is always zero, but the repulsive force term vanishes due to the very large distances between particles in weight space.*

Past works have noted the pathologies of particle-based VI in high dimensions (Zhuo et al., 2018; Ba et al., 2021), but this example is interesting as it does not require an increasing dimensionality. Rather, it is global overparameterization that breaks the asymptotic convergence to LD.

**Analysis of non-free group actions** As we have shown in Example 2.2, Proposition 2.1 already demonstrates some equivalence between (WLD) and (FLD) in the presence of global overparameterization. It can also be combined with Example 2.1 (ii) to construct models exhibiting both local and global overparameterization. Still, we present a more relevant example below, which is a BNN model exhibiting permutational

symmetry. Note that the model will still be different from practical models, in particular because it precludes continuous symmetry. However, it allows for a non-constant NTK, which is an important feature of effective NN models for high-dimensional data (see e.g., Ghorbani et al., 2019; Wei et al., 2019).

**Example 2.3** (simplified BNN model)**.** *Consider the model $f(x; \theta) := \sum_{i=1}^{d} \sin(\theta_i x)$, which is a two-layer BNN with the second layer frozen at initialization.*

*Let the prior support $\operatorname{supp} \pi_\theta$ be contained in $(0, +\infty)^d$. Then by the linear independence of sine functions, for $\mathcal{A}\theta = \mathcal{A}\theta'$ to hold, $\theta'$ must be a permutation of $\theta$, and thus the symmetry in this model can be described by the symmetric group $S_d$ consisting of all permutations on the set $\{1, \ldots, d\}$. The action of $S_n$ on the weight space $\mathbb{R}^d$ is non-free, and the function space is a manifold with boundary, namely a polyhedral cone $C_n := \{\theta \in \mathbb{R}^d : \theta_1 \le \theta_2 \le \cdots \le \theta_d\}$.*

*Let $\{\theta_t\}$ be the trajectory of (WLD) and $p_t$ denote the distribution of $\theta_t$. Appendix B.3 proves that the push-forward distribution $\tilde{p}_t := \mathcal{A}_\# p_t$ follows the Fokker-Planck equation with the Neumann boundary condition:*

$$\begin{cases} \partial_t \tilde{p}_t(\theta) = -\nabla \cdot (\tilde{p}_t(\theta) \nabla_\theta (\log p(\mathbf{Y} \mid \theta, \mathbf{X}) + \log p_\theta(\theta))) + \Delta \tilde{p}_t(\theta), & \theta \in \mathcal{F}^\circ \\ \partial_\theta \tilde{p}_t(\theta) / \partial v = 0, & v \in N_\theta, \theta \in \partial \mathcal{F}, \end{cases} \tag{2}$$

*where $\partial \mathcal{F}$ and $\mathcal{F}^\circ$ are the boundary and the interior of $\mathcal{F}$, respectively, and $N_\theta$ is the set of inward normal vectors of $\mathcal{F}$ at $\theta$. The evolution of $\tilde{p}_t$ is closely related to the* reflected Langevin dynamics *in $\mathcal{F}$ (Sato et al., 2022), which keeps its trajectory in $\mathcal{F}$ by reflecting it at $\partial \mathcal{F}$; when the posterior is strongly log-concave in $C_n$, the connection suggests that the function-space measure $\tilde{p}_t$ may enjoy a fast convergence.[5] In contrast,* convergence of (WLD) to the weight-space posterior may be much slower*, as it will have to visit an exponential number of equivalence classes.*

We note that mixture models exhibit a similar permutational invariance, and their inferential and computational issues have been extensively studied (Celeux et al., 2000; Frühwirth-Schnatter, 2001; Jasra et al., 2005; Frühwirth-Schnatter & Frèuhwirth-Schnatter, 2006). However, those works typically focus on the mixing in the parameter (i.e., weight) space, which is different from our work which only concerns the function space.

## 3 Numerical Study

While our theoretical results have covered two simplified BNN models, the models are still different from those employed in practice. In this section we present numerical experiments that evaluate the efficacy of (WLD)-derived algorithms on practical BNN models. While they cannot provide direct evidence on the equivalence between (WLD) and (FLD), they are still validating a possible consequence of it, as we expect (FLD) to have good convergence properties (when the NTK and the NNGP kernels are not too different, Sec. 2.2) and (WLD) will inherit such a property if the equivalence holds. We will experiment on two setups, a toy 1D regression dataset (Sec. 3.1) and a collection of semi-synthetic datasets adapted from the UCI regression datasets (Sec. 3.2).

We note that it is impossible to implement (WLD) exactly, as it is a continuous-time process; thus, we will experiment with Metropolis-adjusted Langevin algorithm (MALA, Roberts & Stramer, 2002) and unadjusted Langevin algorithm (ULA, Grenander & Miller, 1994), which are two standard, widely-used algorithms derived from LD.[6] More importantly, it is difficult to directly validate the equivalence between (WLD) and (FLD) empirically, as the latter involves the function-space prior which cannot be computed or approximated efficiently; for this reason we have resorted to *indirect experiments*. The experiments also share a similar goal to our theoretical analysis, which is to understand the behavior of (WLD)-derived algorithms on BNN models. In this aspect they complement previous works, by investigating practical, finite-width NN models and eliminating the influence of possible model misspecification in the evaluation.

---

[5]For a bounded convex domain and a smooth boundary, we can prove that (2) describes the density evolution of the reflected LD, and its convergence rate has also been established (Bubeck et al., 2018, Proposition 2.6).

[6]Briefly, ULA simulates a discretization of (WLD) and MALA corrects for the bias arising from the discretization. For simplicity, we may refer to them as simulating (WLD) in the following.

### 3.1 Sample Quality on a Toy Dataset

We first consider BNN inference on a toy 1D regression dataset, and check if the function-space measure induced by simulating (WLD) (i.e., the distribution of $\mathcal{A}\theta_t$) appears to converge at a similar rate, across models with increasing degree of overparameterization.

1. we will visualize the pointwise credible intervals of $\mathcal{A}\theta_t$, which are informative about one-dimensional marginal distributions of the function-space measure;

2. when the training sample size $n$ is small, we approximately evaluate the approximation quality of $(n+1)$-dimensional marginal distributions of $f(\mathbf{X}_e) := (f(x_1), \ldots, f(x_n), f(x_*))$ with $f \sim \pi_{f|\mathcal{D}}$, by estimating the kernelized Stein discrepancy (KSD, Liu et al., 2016; Chwialkowski et al., 2016) between the marginal distribution $q(f(\mathbf{X}_e))$ (where $f = \mathcal{A}\theta_t$ and $\theta_t$ follows (WLD)), and the true marginal posterior $p(f(\mathbf{X}_e))$ (where $f \sim \pi_{f|\mathcal{D}}$).

KSD is often used for measuring sample quality (Gorham & Mackey, 2017; Anastasiou et al., 2023). We use the U-statistic estimator in Liu et al. (2016, Eq. 14), which only requires the specification of a kernel in $\mathbb{R}^{n+1}$, samples from $q$ and the *score function* of $p(f(\mathbf{X}_e))$. Importantly, we can estimate the score since it admits the following decomposition:

$$\nabla_{f(\mathbf{X}_e)} \log p = \nabla_{f(\mathbf{X}_e)} \Big( \log \frac{d\pi_{f(\mathbf{X}_e)}}{d\mu_{Leb}} + \log p(\mathbf{Y} \mid f(\mathbf{X}_e)) \Big)$$

$$= \nabla_{f(\mathbf{X}_e)} \Big( \log \frac{d\pi_{f(\mathbf{X}_e)}}{d\mu_{Leb}} + \log p(\mathbf{Y} \mid f(\mathbf{X})) \Big), \qquad \text{(since } \mathbf{X} \subset \mathbf{X}_e) \qquad (3)$$

where $\pi_{f(\mathbf{X}_e)}$ denotes the respective marginal distribution of $\pi_f$, and $\mu_{Leb}$ denotes the Lebesgue measure. We estimate the first term by fitting nonparametric score estimators (Zhou et al., 2020) on prior samples. The second term can be evaluated in closed form.

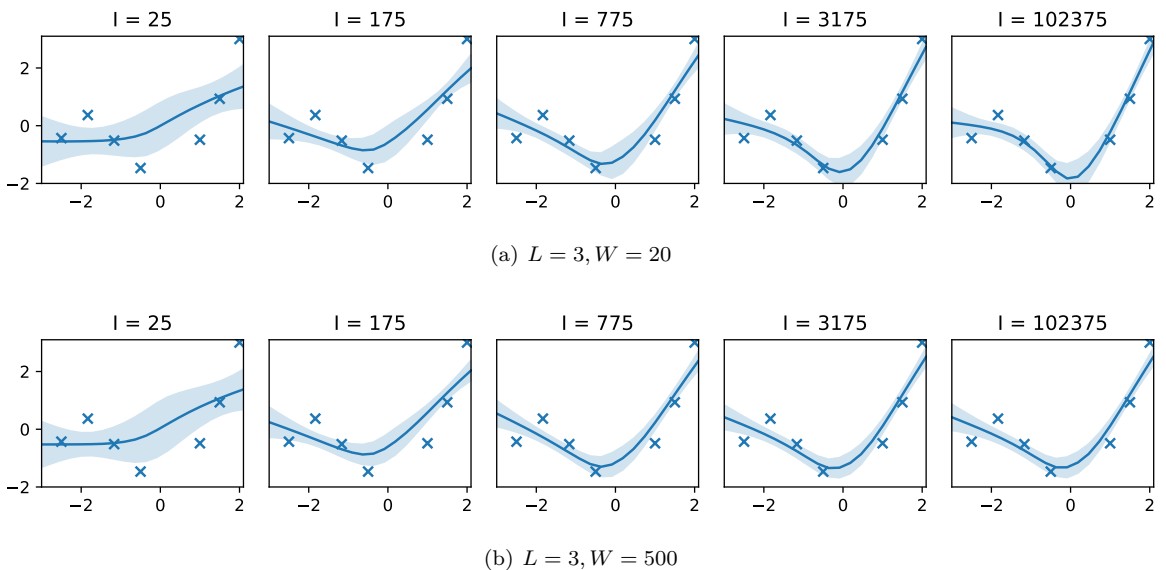

(a) $L = 3, W = 20$

(b) $L = 3, W = 500$

Figure 1: 1D regression: visualization of the induced function-space measure of MALA after $I$ iterations. We plot the pointwise 80% credible intervals. The results for $L = 2$ are deferred to Fig. 4.

We use feed-forward networks with factorized Gaussian priors, and the standard initialization scaling: $f(x; \theta) := f^{(L)}(f^{(L-1)}(\ldots f^{(0)}(x)))$, where

$$f^{(l)}(h^{(l-1)}) := \sigma^{(l)} \Big( B^{(l)} h^{(l-1)} + b^{(l)} \Big), \quad \text{vec}(B^{(l)}) \sim \mathcal{N} \Big( 0, (\dim h^{(l-1)})^{-1} I \Big), \quad b^{(l)} \sim \mathcal{N}(0, 0.2I), \qquad (4)$$

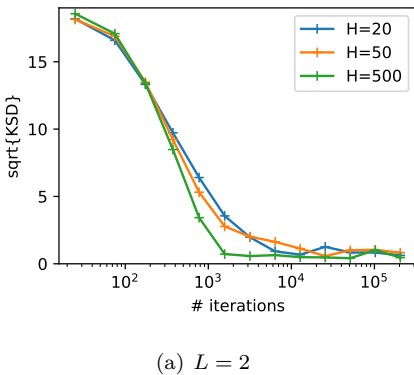

(a) $L = 2$

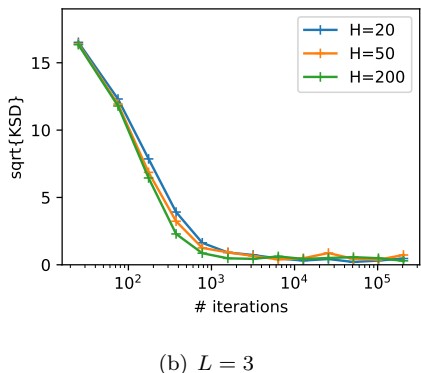

(b) $L = 3$

Figure 2: 1D regression: estimated $\sqrt{\text{KSD}}$ between the LD predictive distribution $q(f(\mathbf{X}_e))$ and the approximate function-space posterior $p(f(\mathbf{X}_e))$. We simulate 1000 LD chains. For the approximate posterior, we estimate the prior score term in (3) using $5 \times 10^6$ samples.

and the activation functions $\sigma^{(l)}$ are SELU (Klambauer et al., 2017) for hidden layers ($l < L$) and the identity map for the output layer ($l = L$). We vary the network depth $L \in \{2, 3\}$, and the width of all hidden layers $W \in [20, 500]$.

The training data is generated as follows: the inputs consist of $\lfloor 2n/3 \rfloor$ evenly spaced points on $[-2.5, -0.5]$, and the remaining points are evenly placed on $[1, 2]$. The output is sampled from $p(y \mid x) = \mathcal{N}(x \sin(1.5x) + 0.125x^3, 0.01)$. We use $n = 7$ for visualization, and $n = 3$ for KSD evaluation. The difference is due to challenges in approximating our KSD: (3) involves score estimation, and in our case we further need the estimate to generalize to out-of-distribution inputs (approximate posterior as opposed to prior samples); both are extremely challenging tasks in high dimensions[7]. We simulate (WLD) with MALA, and evaluate the induced function-space samples for varying number of iterations. The step size is set to $0.025/nW$, so that the function-space updates have a similar scale.

We visualize the posterior approximations in Fig. 1 and Fig. 4, and report the approximate KSD in Fig. 2. As we can see, the convergence appears to happen at a similar rate, which is a possible consequence of the equivalence results.

## 3.2 Average-Case Predictive Performance on Semi-Synthetic Data

The previous experiments cannot scale to larger datasets due to the aforementioned challenges in estimating the KSD. To investigate the behavior of (WLD)-derived algorithms on more realistic datasets, we turn to less direct experiments and check whether *in the absence of model misspecification* (WLD)-derived algorithms will lead to competitive predictive performance.

Our experiments use semi-synthetic datasets adapted from the UCI machine learning repository. Specifically, we modify the UCI datasets by keeping the input data and replacing the output with samples from the model likelihood $p(y \mid x, f_0)$, where $f_0 = f_{\text{BNN}}(\cdot; \theta_0)$ is sampled from the BNN prior:

$$\theta_0 \sim \pi_\theta, \quad y \mid x \sim p(y = \cdot \mid f_{\text{BNN}}(x; \theta_0)). \tag{5}$$

We will consider Gaussian (resp. Laplacian) likelihood and check whether an approximate posterior mean (resp. median) estimator, constructed using the (WLD)-derived algorithms, has a competitive *average-case* performance across randomly sampled $\theta_0$. This will happen if the weight-space algorithms provide a reasonably accurate approximation to the function-space posterior, since the *exact* posterior mean (resp. median) estimator will minimize the similarly defined *average-case* risk

$$\hat{f} \mapsto \mathbb{E}_{f_0 \sim \pi_f} \mathbb{E}_{\mathbf{Y} \sim p(\cdot \mid f_0(\mathbf{X}))} \mathbb{E}_{x_* \sim p_x, y_* \sim p(\cdot \mid f_0(x_*))} \ell(\hat{f}(x_*), y_*), \tag{6}$$

---

[7]Without strong differentiability assumptions, the error of score estimation may suffer from curse of dimensionality (Tsybakov & Zaiats, 2009).

where $\ell$ denotes the loss function derived from the model likelihood, and $\hat{f}$ denotes any estimator that maps the data $(\mathbf{X}, \mathbf{Y})$ to a prediction function (we dropped the dependency on the data for readability). Therefore, competitive predictive performance of the approximate predictor will provide evidence on the quality of posterior approximation. Note that by using a semi-synthetic data generating process, we can allow the input features to have realistic distributions, while avoiding the possible influence of model misspecification which cannot be ruled out in past works that experiment on real-world data (Wenzel et al., 2020; Fortuin et al., 2021).

We estimate the average-case risk (6) for MALA and ULA, using a Monte-Carlo estimate; the full procedure is summarized as Algorithm 1 in appendix. We instantiate (6) with Gaussian and Laplacian likelihoods, which correspond to the square loss and the absolute error loss, respectively. The feed-forward network architecture $f_{\mathrm{BNN}}$ follows Sec. 3.1 by varying $L \in \{2, 3\}, W \in \{50, 200\}$, and we use 80% samples for training and 20% for testing. To understand the performance of the (WLD)-derived algorithms, we report the *Bayes error*, which is the minimum possible average-case risk attainable with *infinite samples*; we also include a baseline that replaces the (WLD)-derived algorithm with an ensemble gradient descent (GD) procedure for a maximum a posterior (MAP) estimate. For all methods, the step size is selected from $\{\eta/2nW : \eta \in \{1, 0.5, 0.1, 0.05, 0.01, 0.005\}\}$ such that the average acceptance rate of the first 200 MALA iterations is closest to 0.7, where $n$ denotes the size of training set.

We plot the Bayes error and the estimated average-case risk against the number of iterations in Fig. 3 and Fig. 5-6 in appendix, and report the performance at the best iteration in Table 1-3. As we can see, across all settings, MALA and ULA lead to a similar predictive performance to GD, and all of them attain errors close to the Bayes error, especially for a dataset with a larger training set. As it is well known that GD methods perform well on DNN models (Du et al., 2018; Allen-Zhu et al., 2019; Mei et al., 2019; Arora et al., 2019a), these results provide further evidence on the efficacy of the (WLD)-derived algorithms.

## 4 Conclusion

In this work we have investigated the function space behavior of weight-space Langevin-type algorithms on overparameterization models. Across multiple settings that encompass simplified BNN models, we have established the equivalence of the function-space pushforward of weight-space LD to its various function-space counterparts. Within their scope, the equivalence results allow us to view weight-space LD as a function-space inference procedure, and understand its behavior by examining the preconditioner in the equivalent function-space dynamics. Numerical experiments provide additional evidence for the efficacy of Langevin-type algorithms on practical feed-forward network models.

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

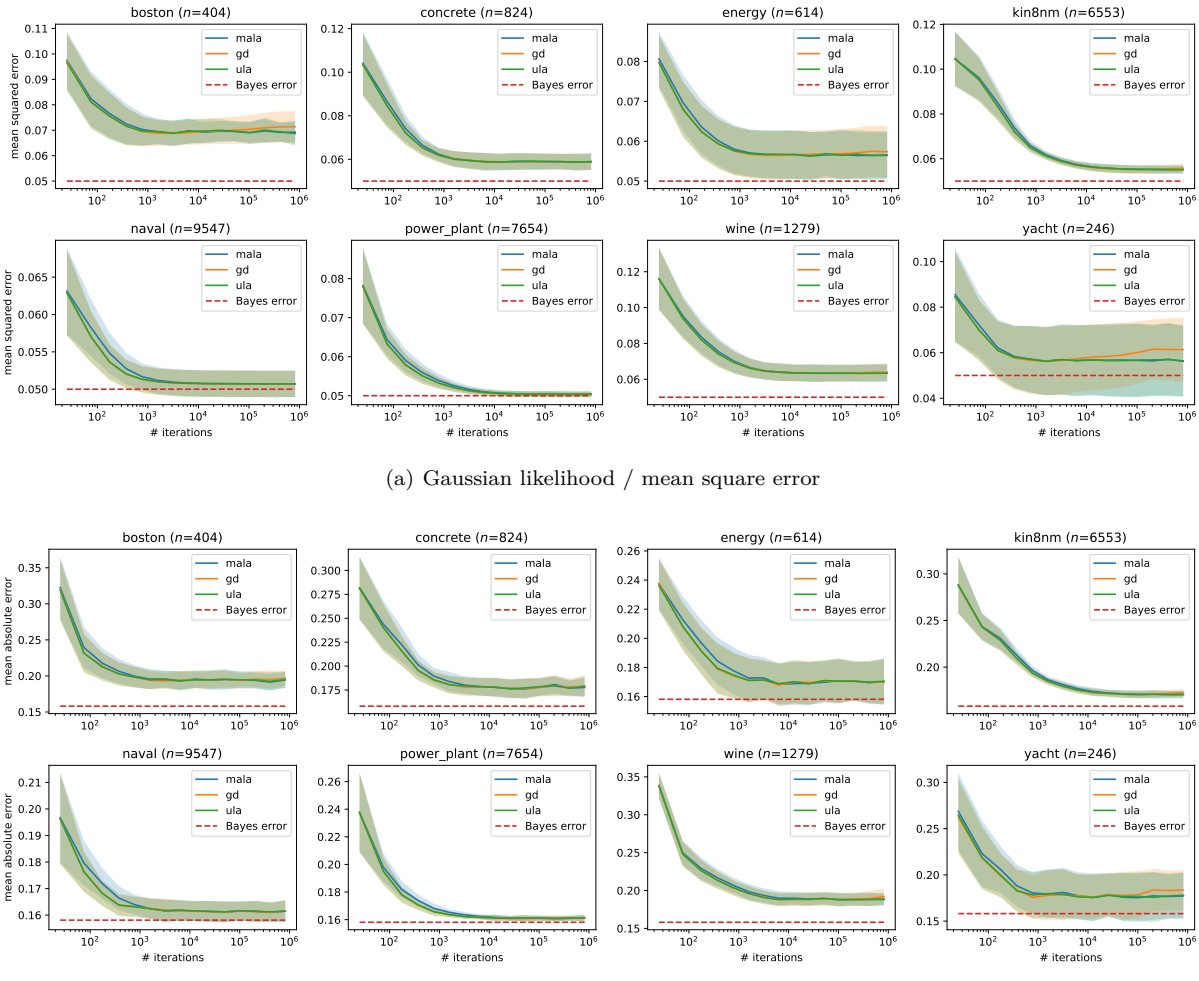

(a) Gaussian likelihood / mean square error

(b) Laplace likelihood / mean absolute error

Figure 3: Semi-synthetic experiment: estimated average-case risk (6) under different choices of likelihood, for $L = 2, W = 200$. Shade indicates standard deviation across 8 independent replications.

Jimmy Ba, Murat A Erdogdu, Marzyeh Ghassemi, Shengyang Sun, Taiji Suzuki, Denny Wu, and Tianzong Zhang. Understanding the variance collapse of SVGD in high dimensions. In *International Conference on Learning Representations*, 2021.

Qinxun Bai, Steven Rosenberg, and Wei Xu. Understanding natural gradient in Sobolev spaces. *arXiv preprint arXiv:2202.06232*, 2022.

George EP Box. Sampling and bayes' inference in scientific modelling and robustness. *Journal of the Royal Statistical Society: Series A (General)*, 143(4):383–404, 1980.

Sébastien Bubeck, Ronen Eldan, and Joseph Lehec. Sampling from a log-concave distribution with projected Langevin Monte Carlo. *Discrete & Computational Geometry*, 59:757–783, 2018.

David R Burt, Sebastian W Ober, Adrià Garriga-Alonso, and Mark van der Wilk. Understanding variational inference in function-space. *arXiv preprint arXiv:2011.09421*, 2020.

José Antonio Carrillo, Katy Craig, and Francesco S Patacchini. A blob method for diffusion. *Calculus of Variations and Partial Differential Equations*, 58(2):1–53, 2019.

Gilles Celeux, Merrilee Hurn, and Christian P Robert. Computational and inferential difficulties with mixture posterior distributions. *Journal of the American Statistical Association*, 95(451):957–970, 2000.

Changyou Chen, Ruiyi Zhang, Wenlin Wang, Bai Li, and Liqun Chen. A unified particle-optimization framework for scalable Bayesian sampling. *arXiv preprint arXiv:1805.11659*, 2018.

Kacper Chwialkowski, Heiko Strathmann, and Arthur Gretton. A kernel test of goodness of fit. In *International conference on machine learning*, pp. 2606–2615. PMLR, 2016.

Manfredo Perdigao Do Carmo. *Riemannian geometry*, volume 6. Springer, 1992.

Simon S Du, Xiyu Zhai, Barnabas Poczos, and Aarti Singh. Gradient descent provably optimizes over-parameterized neural networks. *arXiv preprint arXiv:1810.02054*, 2018.

David Steven Dummit and Richard M Foote. *Abstract algebra*, volume 3. Wiley Hoboken, 2004.

Vincent Fortuin, Adrià Garriga-Alonso, Florian Wenzel, Gunnar Rätsch, Richard Turner, Mark van der Wilk, and Laurence Aitchison. Bayesian neural network priors revisited. *arXiv preprint arXiv:2102.06571*, 2021.

Sylvia Frühwirth-Schnatter. Markov chain Monte Carlo estimation of classical and dynamic switching and mixture models. *Journal of the American Statistical Association*, 96(453):194–209, 2001.

Sylvia Frühwirth-Schnatter and Sylvia Frèuhwirth-Schnatter. *Finite mixture and Markov switching models*, volume 425. Springer, 2006.

Behrooz Ghorbani, Song Mei, Theodor Misiakiewicz, and Andrea Montanari. Limitations of lazy training of two-layers neural networks. *arXiv preprint arXiv:1906.08899*, 2019.

Mark Girolami and Ben Calderhead. Riemann manifold Langevin and Hamiltonian Monte Carlo methods. *Journal of the Royal Statistical Society Series B: Statistical Methodology*, 73(2):123–214, 2011.

Jackson Gorham and Lester Mackey. Measuring sample quality with kernels. In *International Conference on Machine Learning*, pp. 1292–1301. PMLR, 2017.

Ulf Grenander and Michael I Miller. Representations of knowledge in complex systems. *Journal of the Royal Statistical Society: Series B (Methodological)*, 56(4):549–581, 1994.

Brian C Hall. *Lie groups, Lie algebras, and representations*. Springer, 2013.

Soufiane Hayou, Arnaud Doucet, and Judith Rousseau. Exact convergence rates of the neural tangent kernel in the large depth limit. *arXiv e-prints*, pp. arXiv–1905, 2019.

Bobby He, Balaji Lakshminarayanan, and Yee Whye Teh. Bayesian deep ensembles via the neural tangent kernel. *Advances in Neural Information Processing Systems*, 33:1010–1022, 2020.

Pavel Izmailov, Sharad Vikram, Matthew D Hoffman, and Andrew Gordon Gordon Wilson. What are Bayesian neural network posteriors really like? In *International Conference on Machine Learning*, pp. 4629–4640. PMLR, 2021.

Arthur Jacot, Franck Gabriel, and Clément Hongler. Neural tangent kernel: Convergence and generalization in neural networks. *Advances in Neural Information Processing Systems*, 31, 2018.

Ajay Jasra, Chris C Holmes, and David A Stephens. Markov chain monte carlo methods and the label switching problem in bayesian mixture modeling. 2005.

Pauwels JE. Riemannian submersions of Brownian motions. *Stochastics: An International Journal of Probability and Stochastic Processes*, 29(4):425–436, 1990.

Günter Klambauer, Thomas Unterthiner, Andreas Mayr, and Sepp Hochreiter. Self-normalizing neural networks. *Advances in Neural Information Processing Systems*, 30, 2017.

Richard Kurle, Ralf Herbrich, Tim Januschowski, Yuyang Bernie Wang, and Jan Gasthaus. On the detrimental effect of invariances in the likelihood for variational inference. *Advances in Neural Information Processing Systems*, 35:4531–4542, 2022.

Jaehoon Lee, Jascha Sohl-dickstein, Jeffrey Pennington, Roman Novak, Sam Schoenholz, and Yasaman Bahri. Deep neural networks as Gaussian processes. In *International Conference on Learning Representations*, 2018.

Jaehoon Lee, Lechao Xiao, Samuel Schoenholz, Yasaman Bahri, Roman Novak, Jascha Sohl-Dickstein, and Jeffrey Pennington. Wide neural networks of any depth evolve as linear models under gradient descent. *Advances in Neural Information Processing Systems*, 32, 2019.

John M. Lee. *Introduction to Smooth Manifolds*. Springer, 2012.

John M Lee. *Introduction to Riemannian manifolds*. Springer, 2018.

Qiang Liu, Jason Lee, and Michael Jordan. A kernelized stein discrepancy for goodness-of-fit tests. In *International conference on machine learning*, pp. 276–284. PMLR, 2016.

Chao Ma and José Miguel Hernández-Lobato. Functional variational inference based on stochastic process generators. *Advances in Neural Information Processing Systems*, 34:21795–21807, 2021.

Chao Ma, Yingzhen Li, and José Miguel Hernández-Lobato. Variational implicit processes. In *International Conference on Machine Learning*, pp. 4222–4233. PMLR, 2019.

Alexander G de G Matthews, Mark Rowland, Jiri Hron, Richard E Turner, and Zoubin Ghahramani. Gaussian process behaviour in wide deep neural networks. *arXiv preprint arXiv:1804.11271*, 2018.

Song Mei, Theodor Misiakiewicz, and Andrea Montanari. Mean-field theory of two-layers neural networks: dimension-free bounds and kernel limit. In *Conference on Learning Theory*, pp. 2388–2464. PMLR, 2019.

Ankur Moitra and Andrej Risteski. Fast convergence for Langevin diffusion with manifold structure, September 2020. arXiv:2002.05576 [cs, math, stat].

Ian Osband, John Aslanides, and Albin Cassirer. Randomized prior functions for deep reinforcement learning. *Advances in Neural Information Processing Systems*, 31, 2018.

Omiros Papaspiliopoulos, Gareth O Roberts, and Martin Sköld. A general framework for the parametrization of hierarchical models. *Statistical Science*, pp. 59–73, 2007.

Gareth O Roberts and Osnat Stramer. Langevin diffusions and Metropolis-Hastings algorithms. *Methodology and computing in applied probability*, 4:337–357, 2002.

Gareth O Roberts and Richard L Tweedie. Exponential convergence of Langevin distributions and their discrete approximations. *Bernoulli*, pp. 341–363, 1996.

Ichirô Satake. On a generalization of the notion of manifold. *Proceedings of the National Academy of Sciences*, 42(6):359–363, 1956.

Kanji Sato, Akiko Takeda, Reiichiro Kawai, and Taiji Suzuki. Convergence error analysis of reflected gradient Langevin dynamics for globally optimizing non-convex constrained problems. *arXiv preprint arXiv:2203.10215*, 2022.

Samuel S Schoenholz, Justin Gilmer, Surya Ganguli, and Jascha Sohl-Dickstein. Deep information propagation. *arXiv preprint arXiv:1611.01232*, 2016.

Jiaxin Shi, Mohammad Emtiyaz Khan, and Jun Zhu. Scalable training of inference networks for Gaussian-process models. In *International Conference on Machine Learning*, pp. 5758–5768. PMLR, 2019.

Shengyang Sun, Guodong Zhang, Jiaxin Shi, and Roger Grosse. Functional variational Bayesian neural networks. *arXiv preprint arXiv:1903.05779*, 2019.

A B Tsybakov and Vladimir Zaiats. *Introduction to Nonparametric Estimation*. Springer series in statistics. Springer, New York, NY, December 2009.

Cédric Villani. *Optimal transport: old and new*, volume 338. Springer, 2009.

Ziyu Wang, Tongzheng Ren, Jun Zhu, and Bo Zhang. Function space particle optimization for Bayesian neural networks. In *International Conference on Learning Representations*, 2019.

Colin Wei, Jason D Lee, Qiang Liu, and Tengyu Ma. Regularization matters: Generalization and optimization of neural nets v.s. their induced kernel. In H. Wallach, H. Larochelle, A. Beygelzimer, F. d'Alché Buc, E. Fox, and R. Garnett (eds.), *Advances in Neural Information Processing Systems*, volume 32, 2019.

Florian Wenzel, Kevin Roth, Bastiaan S Veeling, Jakub Świkatkowski, Linh Tran, Stephan Mandt, Jasper Snoek, Tim Salimans, Rodolphe Jenatton, and Sebastian Nowozin. How good is the Bayes posterior in deep neural networks really? *arXiv preprint arXiv:2002.02405*, 2020.

Blake Woodworth, Suriya Gunasekar, Jason D Lee, Edward Moroshko, Pedro Savarese, Itay Golan, Daniel Soudry, and Nathan Srebro. Kernel and rich regimes in overparametrized models. In *Conference on Learning Theory*, pp. 3635–3673. PMLR, 2020.

Greg Yang. Scaling limits of wide neural networks with weight sharing: Gaussian process behavior, gradient independence, and neural tangent kernel derivation. *arXiv preprint arXiv:1902.04760*, 2019.

Yaming Yu and Xiao-Li Meng. To center or not to center: That is not the question—an ancillarity–sufficiency interweaving strategy (asis) for boosting mcmc efficiency. *Journal of Computational and Graphical Statistics*, 20(3):531–570, 2011.

Ruqi Zhang, Chunyuan Li, Jianyi Zhang, Changyou Chen, and Andrew Gordon Wilson. Cyclical stochastic gradient mcmc for Bayesian deep learning. *arXiv preprint arXiv:1902.03932*, 2019.

Yuhao Zhou, Jiaxin Shi, and Jun Zhu. Nonparametric score estimators. In *International Conference on Machine Learning*, pp. 11513–11522. PMLR, 2020.

Jingwei Zhuo, Chang Liu, Jiaxin Shi, Jun Zhu, Ning Chen, and Bo Zhang. Message passing Stein variational gradient descent. In *International Conference on Machine Learning*, pp. 6018–6027. PMLR, 2018.

## A    Background Knowledge

### A.1    Groups

A *group $H$* is a set equipped with an operation $\Psi : H \times H \to H$. It satisfies the following properties:

- There is a unit element $e$ in $H$ such that $\Psi(e, \varphi) = \Psi(\varphi, e) = \varphi$ for every $\varphi \in H$.

- Every element $\varphi \in H$ has an inverse $\varphi^{-1} \in H$ such that $\Psi(\varphi, \varphi^{-1}) = \Psi(\varphi^{-1}, \varphi) = e$.

- The operation $\Psi$ is associative, i.e., for $\varphi_1, \varphi_2, \varphi_3 \in H$, it holds that $\Psi(\varphi_1, \Psi(\varphi_2, \varphi_3)) = \Psi(\Psi(\varphi_1, \varphi_2), \varphi_3)$.

For simplicity, we use $\varphi_1 \varphi_2$ to denote $\Psi(\varphi_1, \varphi_2)$ for every $\varphi_1, \varphi_2 \in H$.

A group $H$ is a *Lie group* if it is a smooth manifold (Hall, 2013), and we say $H$ is *discrete* if for each $\varphi \in H$ there exists a neighborhood $U_\varphi \ni \varphi$ containing only $\varphi$.

### A.2    Riemannian Manifolds

To aid understanding, we provide a conceptual introduction of some relevant notions in Riemannian geometry in this section. We refer interested readers to Do Carmo (1992); Lee (2018) for rigorous definitions.

A *$k$-dimensional manifold $\mathcal{M}$* is a topological space locally resembling a $k$-dimensional Euclidean space. Specifically, for any $p \in \mathcal{M}$, there exist open neighborhoods $\mathcal{M} \supset U \ni p$ and $\mathbb{R}^k \supset V \ni 0$ and a homeomorphism $\psi : V \to U$. The map $\psi$ is called a *coordinate map near $p$* if $\psi(0) = p$, and the *coordinate* of a point $q \in U$ is $\psi^{-1}(q)$. If two coordinate maps $\psi_1, \psi_2$ overlap, we require the transition $\psi_2^{-1} \circ \psi_1$ to be smooth.

The *tangent space* at $p \in \mathcal{M}$ is a $k$-dimensional linear space "orthogonal to $\mathcal{M}$", denoted as $T_p \mathcal{M}$. A *Riemannian structure* equips the tangent space $T_p \mathcal{M}$ with an inner product $\langle \cdot, \cdot \rangle_p$ for every $p \in \mathcal{M}$. Given a coordinate map $\psi : V \to U$ and $a \in V$, the differential $\mathrm{d}\psi|_a$ at $a$ is a linear bijection between $\mathbb{R}^k$ and $T_{\psi(a)} \mathcal{M}$. Let $e_1, e_2, \ldots, e_k \in \mathbb{R}^k$ be the standard basis of $\mathbb{R}^k$, then $\{E_i(a) := (\mathrm{d}\psi|_a)(e_i)\}_{i \in [k]}$ is a basis of $T_{\psi(a)} \mathcal{M}$. We call $g_{ij}(a) := \langle E_i(a), E_j(a) \rangle_{\psi(a)}$ the *coordinate representation of the Riemannian metric* and $G(a) := (g_{ij}(a))_{i,j \in [k]} \in \mathbb{R}^{k \times k}$ the *coordinate representation of the metric matrix*. For simplicity, we omit the dependence on $a$ of $g_{ij}(a)$ and $G(a)$.

Also, under a coordinate $\psi : V \to U$, the *volume* of a set $R \subset U$ is defined as (Do Carmo, 1992, p. 45)

$$\mathrm{Vol}(R) := \int_{\psi^{-1}(R)} \sqrt{|G|} \mathrm{d}\mu_{Leb},$$

where $|G|$ denotes the determinant of $G$, and $\mu_{Leb}$ is the $k$-dimensional Lebesgue measure. The measure $\mathrm{d}\mu_{\mathcal{M}} := \sqrt{|G|} \mathrm{d}\mu_{Leb}$ is called the *Riemannian measure* or the *volume form*, and is independent of the choice of the coordinate $\psi$.

**Example A.1.** *As an example, let $\mathcal{M} := \mathbb{T}^2 := \{(\cos \alpha, \sin \alpha, \cos \beta, \sin \beta) : \alpha, \beta \in [0, 2\pi)\}$ be the two-dimensional torus considered in Example 2.2. For any $p = (\cos \alpha, \sin \alpha, \cos \beta, \sin \beta) \in \mathbb{T}^2$, we can find a coordinate map $\psi(\zeta, \xi) := (\cos(\alpha + \zeta), \sin(\alpha + \zeta), \cos(\beta + \xi), \sin(\beta + \xi)) \in \mathbb{R}^4$ with $(\zeta, \xi) \in V := (-1, 1) \times (-1, 1)$. The tangent space $T_p \mathbb{T}^2 = p + \{(-t \sin \alpha, t \cos \alpha, -s \sin \beta, s \cos \beta) : t, s \in \mathbb{R}\}$ is the plane orthogonal to $p$. For $t_1, t_2, s_1, s_2 \in \mathbb{R}$ and tangent vectors $v_1 = p + (-t_1 \sin \alpha, t_1 \cos \alpha, -s_1 \sin \beta, s_1 \cos \beta)$ and $v_2 = p + (-t_2 \sin \alpha, t_2 \cos \alpha, -s_2 \sin \beta, s_2 \cos \beta)$, we can define an inner product $\langle v_1, v_2 \rangle_p := (v_1 - p)^\top (v_2 - p) = t_1 t_2 + s_1 s_2$. The differential of $\psi$ is $(\mathrm{d}\psi|_0)(t, s) = (-t \sin \alpha, t \cos \alpha, -s \sin \beta, s \cos \beta)$, mapping the basis $e_1 = (1, 0), e_2 = (0, 1)$ in $\mathbb{R}^2$ to $E_1 = p + (-\sin \alpha, \cos \alpha, 0, 0), E_2 = p + (0, 0, -\sin \beta, \cos \beta) \in T_p \mathbb{T}^2$. The coordinate representation of the metric is such that $g_{ij}(0) = 1$ if $i = j$ and $g_{ij}(0) = 0$ otherwise, and the metric matrix is $G(0) = \begin{pmatrix} 1 & 0 \\ 0 & 1 \end{pmatrix}$.*

## B   Proofs

### B.1   Details in Example 2.1

As mentioned in this example, when the input space $\mathcal{X}$ has finite cardinality $K \in \mathbb{N}$, a function $f : \mathcal{X} \to \mathbb{R}$ can be identified as the vector $(f(\mathbf{x}_1), f(\mathbf{x}_2), \ldots, f(\mathbf{x}_K)) \in \mathbb{R}^K$. Conversely, for a vector $v = (v_1, v_2, \ldots, v_K) \in \mathbb{R}^K$ we can define a function $f_v : \mathcal{X} \to \mathbb{R}$ such that $f_v(\mathbf{x}_j) = v_j$ for $j = 1, \ldots, K$. In the remaining part, we will represent the function $f_v \in \mathbb{R}^{\mathcal{X}}$ by the vector $v \in \mathbb{R}^K$.

*Proof of Example 2.1.* Since the map $\mathcal{A}$ is linear, there exists a matrix $A \in \mathbb{R}^{K \times d}$ such that $\mathcal{A}\theta = A\theta$ for every $\theta \in \mathbb{R}^d$.

Now the claim (i) follows by Proposition 2.1 (b): clearly, in this case $H = \{e\}$ fulfills the conditions in the proposition.

We now turn to (ii). When $\mathcal{A}$ is a surjection with $d > K$. Let $\mathrm{Ker}(\mathcal{A}) := \mathrm{Ker}(A) := \{x \in \mathbb{R}^d : Ax = 0\}$ be the kernel of $A$, and $\mathrm{Ran}(\mathcal{A}^{\top}) := \mathrm{Ran}(A^{\top}) := \{A^{\top}y \in \mathbb{R}^d : y \in \mathbb{R}^K\}$ be the range of $A^{\top}$, then the space $\mathbb{R}^d$ has the orthogonal decomposition $\mathbb{R}^d = \mathrm{Ker}(A) \oplus \mathrm{Ran}(A^{\top})$, and there exists an orthogonal matrix $Q \in \mathbb{R}^{d \times d}$ such that $Q(\mathrm{Ker}(A)) = \{0\}^k \times \mathbb{R}^{d-k}$ and $Q(\mathrm{Ran}(A^{\top})) = \mathbb{R}^k \times \{0\}^{d-k}$, where $k := \dim \mathrm{Ran}(A^{\top}) \leq K < d$. Then, $\theta \in \mathbb{R}^d$ has the representation $\theta = Q^{\top}(\theta^{\|}, \theta^{\perp})$ for $\theta^{\|} \in \mathbb{R}^k$ and $\theta^{\perp} \in \mathbb{R}^{d-k}$. Under this representation, $\mathcal{A}$ becomes $\tilde{\mathcal{A}} := \mathcal{A} \circ Q^{\top}$ and $\tilde{\mathcal{A}}|_{\mathbb{R}^k \times \{0\}^{d-k}}$ is a bijection. We can also define a reduced likelihood $\tilde{p}(\mathbf{Y} \mid \theta^{\|}, \mathbf{X}) := p(\mathbf{Y} \mid Q^{\top}(\theta^{\|}, 0), \mathbf{X})$, then

$$\mathrm{d}\begin{pmatrix} \theta_t^{\|} \\ \theta_t^{\perp} \end{pmatrix} = \mathrm{d}(Q\theta) = (Q\nabla_\theta \log p(\mathbf{Y} \mid \theta_t, \mathbf{X}) - Q\theta_t)\mathrm{d}t + \sqrt{2}Q\mathrm{d}B_t \quad (p_\theta = \mathcal{N}(0, I_d))$$

$$= \left( \begin{pmatrix} \nabla_{\theta^{\|}} \log \tilde{p}(\mathbf{Y} \mid \theta_t^{\|}, \mathbf{X}) \\ 0 \end{pmatrix} - \begin{pmatrix} \theta_t^{\|} \\ \theta_t^{\perp} \end{pmatrix} \right)\mathrm{d}t + \sqrt{2}\mathrm{d}\tilde{B}_t \qquad (Q\mathrm{d}B_t \overset{d}{=} \sqrt{QQ^{\top}}\mathrm{d}B_t = \mathrm{d}\tilde{B}_t)$$

where $\tilde{B}_t$ denotes another Brownian motion. Therefore, $\theta_t^{\|}$ and $\theta_t^{\perp}$ factorize to independent processes, and the equivalence holds by applying the result of (i) to $\theta_t^{\|}$ and $\tilde{\mathcal{A}}$ (and noting that $\tilde{\mathcal{A}}\theta_t^{\|} = \mathcal{A}\theta_t$). $\qquad \square$

### B.2   Proof of Proposition 2.1

*Proof of Proposition 2.1.* By definitions, for any $f \in \mathcal{F}$, there exists some $\theta \in \mathbb{R}^d$ and one of its neighborhood $N$ such that $f = \mathcal{A}\theta$, and that for $U = \mathcal{A}(N)$, $(U, \mathcal{A}|_N)$ forms a coordinate chart. On this chart, the coordinate matrix of the pushforward metric tensor equals identity, by its definition. Thus, the coordinate representation (FLD) reduces to

$$\mathrm{d}\theta_t = \nabla_\theta \left( \log p(\mathbf{Y} \mid \theta_t, \mathbf{X}) + \log \frac{d\pi_f}{d\mu_{\mathcal{F}}} \right)\mathrm{d}t + \sqrt{2}\mathrm{d}B_t,$$

and it differs from (WLD) only on the prior term. When condition (a) in the proposition holds, the prior is uniform so the gradient vanishes. When condition (b) holds, the group is trivial and the quotient map $\mathcal{A}$ is a bijection. Thus, it suffices to show that for all $\theta \in \mathrm{supp}\,\pi_\theta$, we have

$$\frac{d\pi_f}{d\mu_{\mathcal{F}}}(\mathcal{A}\theta) = \frac{d\pi_\theta}{d\mu_{Leb}}(\theta) = p_\theta(\theta),$$

where $\mu_{Leb}$ denotes the Lebesgue measure. By the change of measure formula, the above will be implied by

$$\pi_f \overset{(i)}{=} \mathcal{A}_{\#}\pi_\theta, \quad \mu_{\mathcal{F}} \overset{(ii)}{=} \mathcal{A}_{\#}\mu_{Leb}.$$

(i) is the definition of $\pi_f$. For (ii), let $\zeta : \mathcal{F} \to \mathbb{R}$ be any measurable function with a compact support, $\{(U_i = \mathcal{A}(N_i), \mathcal{A}|_{N_i}) : i \in [h]\}$ be a finite chart covering of $\mathrm{supp}\,\zeta$, and $\{\rho_i\}$ be a corresponding partition of unity. Then

$$\int_{\mathcal{F}} \zeta(f)\mu_{\mathcal{F}}(\mathrm{d}f) = \sum_{i=1}^h \int_{N_i} (\rho_i\zeta)(\mathcal{A}(\theta))\sqrt{|G(\theta)|}\mu_{Leb}(\mathrm{d}\theta) = \int_{\mathcal{A}^{-1}(\mathrm{supp}\,\zeta)} \zeta(\mathcal{A}(\theta))\mu_{Leb}(\mathrm{d}\theta).$$

This establishes (ii), and thus completes the proof. □

### B.3 Details in Example 2.3

Recall the definition of the cone $C_d := \{x \in \mathbb{R}^d : x_1 \leq x_2 \leq ... \leq x_d\}$, and the group $S_d$ that consists of all permutations of length $d$. An action of $S_d$ on $\mathbb{R}^d$ can be naturally defined, under which we have $C_d = \mathbb{R}^d/S_d$.

We introduce a few additional notations. For $x \in \mathbb{R}^d$, the *stabilizer subgroup* is defined as $\text{Stab}_{S_d} x := \{\varphi \in S_d : \varphi \cdot x = x\}$, and the orbit is $S_d \cdot x := \{\varphi \cdot x : \varphi \in S_d\}$. A vector $n_x \in \mathbb{R}^d$ is an *inward normal vector* of $C_d$ at $x$ if $\langle n_x, y - x \rangle \geq 0$ holds for all $y \in C_d$. Denote by $N_x$ the set of all inward normal vector of $C_d$ at $x$. For any $f : \mathbb{R}^d \to \mathbb{R}$, define the function

$$\tilde{f} : C_d \to \mathbb{R}, \quad \tilde{f}(x) := \frac{1}{|S_d|} \sum_{\varphi \in S_d} f(\varphi \cdot x). \tag{7}$$

When $f$ is the density function of a measure $\pi$ on $\mathbb{R}^d$, the pushforward measure under the quotient map $\mathbb{R}^d \to C_d$ has the density function $\tilde{f}$. The following lemma shows that the directional derivative of $\tilde{f}$ along the normal direction vanishes.

**Lemma B.1.** *Let $x \in C_d$ and assume $f$ is differentiable at every $y \in S_d \cdot x$. Then*

$$D_v \tilde{f}(x) = \frac{1}{|S_d|} \sum_{y := \psi \cdot x \in S_d \cdot x} D_{\psi \cdot W_x(v)} f(y), \quad \text{where} \quad W_x(v) := \sum_{\varphi \in \text{Stab } x} \varphi \cdot v,$$

*where $D_v$ denotes the directional derivative along $v$. Moreover, $W_x(v) = v$ for $x \in C_n^\circ$ and $v \in \mathbb{R}^d$, and $W_x(v) = 0$ for $x \in \partial C_d$ and $v \in N_x$.*

We postpone the proof of the above lemma to the end of this section, and first present the following lemma, which implies the invariance of the Fokker-Planck equation under orthogonal transformations.

**Lemma B.2.** *Let $f, g : \mathbb{R}^d \to \mathbb{R}$ be two functions and $Q \in \mathbb{R}^{d \times d}$ be an orthogonal matrix, then $[\nabla(f \circ Q)]^T \nabla(g \circ Q) = [(\nabla f)^T \nabla g] \circ Q$ and $\Delta(f \circ Q) = \Delta f \circ Q$, in which $Q$ is also regarded as a linear map $Q : \mathbb{R}^d \to \mathbb{R}^d$.*

*Proof.* Note that $\nabla(f \circ Q) = Q^T(\nabla f \circ Q)$. Let $Q_i$ be the $i$-th column of $Q$, then

$$[\nabla(f \circ Q)]^T \nabla(g \circ Q) = \sum_{i=1}^d (\nabla f \circ Q)^T Q_i Q_i^T (\nabla g \circ Q) = (\nabla f \circ Q)^T (\nabla g \circ Q).$$

A similar result also holds for the Laplacian:

$$\Delta(f \circ Q) = \sum_{i=1}^d \partial_i \partial_i (f \circ Q) = \sum_{i,j=1}^d \partial_i (\partial_j f \circ Q) q_{ji} = \sum_{i,j,k=1}^d (\partial_k \partial_j f \circ Q) q_{ji} q_{ki}.$$

As $Q$ is orthogonal, we know $\sum_{i=1}^d q_{ji} q_{ki} = \delta_{jk}$, which completes the proof. □

As the pushforward measure $\mathcal{A}_{\#} p$ has density $\tilde{p}$, the following proposition establishes the equivalence result claimed in the text.

**Proposition B.1.** *Let $p : \mathbb{R}^d \to \mathbb{R}$ be any function that is invariant under the action of $S_d$, and $X_t$ follow the Langevin dynamics on $\mathbb{R}^d$,*

$$dX_t = \nabla \log p(X_t)dt + \sqrt{2}dB_t.$$

*Then, the pushforward density $\tilde{p}_t$ of $X_t$ will evolve as*

$$\begin{cases} \partial_t \tilde{p}_t = -\nabla \cdot (\tilde{p}_t \nabla \log p) + \Delta \tilde{p}_t, & \text{in } C_n^\circ, \\ \frac{\partial \tilde{p}_t}{\partial v}(x) = 0, & \forall v \in N_x, x \in \partial C_d. \end{cases}$$

*Proof.* Let $p_t$ be the density of the distribution of $X_t$, then it follows the Fokker-Planck equation

$$\partial_t p_t = -\nabla \cdot (p_t \nabla \log p) + \Delta p_t = -(\nabla p_t)^T \nabla \log p - p_t \Delta \log p + \Delta p_t.$$

For $\varphi \in S_d$, we denote $P_\varphi \in \mathbb{R}^{d \times d}$ by the corresponding matrix such that $\varphi \cdot x = P_\varphi x$ for every $x \in \mathbb{R}^d$. Then, $P_\varphi$ is an orthogonal matrix, and by Lemma B.2

$$\partial_t(p_t \circ P_\varphi) = (\partial_t p_t) \circ P_\varphi = -(\nabla p_t \cdot \nabla \log p) \circ P_\varphi - (p_t \Delta \log p) \circ P_\varphi + (\Delta p_t) \circ P_\varphi$$
$$= -[\nabla(p_t \circ P_\varphi)]^T \nabla \log p - (p_t \circ P_\varphi) \Delta \log p + \Delta(p_t \circ P_\varphi),$$

where the first equation is because $P_\varphi$ is independent to $t$, and the last equation follows from Lemma B.2 and $\log p \circ P_\varphi = \log p$.

Therefore, we obtain the equation for $\tilde{p}_t$:

$$\partial_t \tilde{p}_t = \frac{1}{|S_d|} \sum_{\varphi \in S_d} \partial(p_t \circ P_\varphi) = \frac{1}{|S_d|} \sum_{\varphi \in S_d} \left( -\nabla \cdot ((p_t \circ P_\varphi) \nabla \log p) + \Delta(p_t \circ P_\varphi) \right)$$
$$= -\nabla \cdot (\tilde{p}_t \nabla \log p) + \Delta \tilde{p}_t. \tag{8}$$

Combining with Lemma B.1 yields the boundary condition

$$\frac{\partial \tilde{p}_t}{\partial v}(x) = 0, \quad \forall v \in N_x, x \in \partial C_d. \tag{9}$$

$\square$

*Proof of Lemma B.1.* Since the group action is linear (i.e., $\varphi \cdot (x+y) = \varphi \cdot x + \varphi \cdot y$ and $\varphi \cdot (tx) = t\varphi \cdot x$), we have

$$D_v \tilde{f}(x) = \lim_{t \to 0+} \frac{1}{t} \left( \tilde{f}(x+tv) - \tilde{f}(x) \right) = \frac{1}{|S_d|} \sum_{\varphi \in S_d} D_{\varphi \cdot v} f(\varphi \cdot x).$$

To simplify the above summation, we introduce the coset $\varphi \operatorname{Stab} x := \{\varphi\psi : \psi \in \operatorname{Stab} x\}$ for each $\varphi \in S_d$, and the set of cosets $S_d / \operatorname{Stab} x := \{\varphi \operatorname{Stab} x : \varphi \in S_d\}$. Clearly, any two cosets are either equal or disjoint, and the group $S_d$ is partitioned by $S_d / \operatorname{Stab} x$. The orbit-stabilizer theorem (Dummit & Foote, 2004, p. 114) states that the map $\varphi \operatorname{Stab} x \mapsto \varphi \cdot x$ is a bijection between cosets $S_d / \operatorname{Stab} x$ and the orbit $S_d \cdot x$, and thus[8]

$$D_v \tilde{f}(x) = \frac{1}{|S_d|} \sum_{\varphi \in S_d} D_{\varphi \cdot v} f(\varphi \cdot x)$$
$$= \frac{1}{|S_d|} \sum_{\substack{\varphi \in C \\ C = \psi \operatorname{Stab} x \in S_d / \operatorname{Stab} x}} D_{\varphi \cdot v} f(\varphi \cdot x) \quad \text{(partition)}$$
$$= \frac{1}{|S_d|} \sum_{\substack{\varphi \in \psi \operatorname{Stab} x \\ y := \psi \cdot x \in S_d \cdot x}} D_{\varphi \cdot v} f(\varphi \cdot x) \quad (\psi \operatorname{Stab} x \mapsto \psi \cdot x \text{ bijective})$$
$$= \frac{1}{|S_d|} \sum_{y := \psi \cdot x \in S_d \cdot x} \sum_{\varphi' \in \operatorname{Stab} x} D_{\psi \cdot (\varphi' \cdot v)} f(y) \quad (\varphi' := \psi^{-1} \varphi)$$
$$= \frac{1}{|S_d|} \sum_{y := \psi \cdot x \in S_d \cdot x} D_{\psi \cdot W_x(v)} f(y). \quad \text{(linearity of } D_{(\cdot)} f)$$

This proves the first claim.

For any interior point $x \in C_d^\circ$, we have $\operatorname{Stab} x = \{e\}$ and thus $W_x(v) = v$. For any boundary point $x \in \partial C_d$, the stabilizer subgroup is non-trivial, and it remains to show that $W_x(v) = 0$ for normal vectors.

---

[8]It can be verified that the proof is independent on the choice of $\psi$.

Table 1: Semi-synthetic experiment: average-case test risk for the best stopping iteration, for $L = 2, W = 200$.

| Likelihood | Algorithm | boston | concrete | energy | kin8nm | naval | power plant | wine | yacht |
|---|---|---|---|---|---|---|---|---|---|
| Gaussian | MALA | 0.067 | 0.058 | 0.056 | 0.055 | 0.051 | 0.050 | 0.063 | 0.055 |
|  | GD | 0.068 | 0.059 | 0.056 | 0.055 | 0.051 | 0.050 | 0.063 | 0.056 |
|  | ULA | 0.067 | 0.058 | 0.056 | 0.055 | 0.051 | 0.050 | 0.063 | 0.055 |
| Laplacian | MALA | 0.188 | 0.174 | 0.167 | 0.170 | 0.160 | 0.160 | 0.186 | 0.172 |
|  | GD | 0.189 | 0.175 | 0.167 | 0.170 | 0.161 | 0.160 | 0.186 | 0.173 |
|  | ULA | 0.188 | 0.175 | 0.167 | 0.170 | 0.161 | 0.160 | 0.186 | 0.171 |

An element $\varphi \in S_d$ can be identified as a permutation matrix $P_\varphi \in \mathbb{R}^{d \times d}$ s.t. the group action is the matrix-vector multiplication $\varphi \cdot v = P_\varphi v$, and clearly, the stabilizer of $x \in \partial C_d$ always has the form of a Cartesian product, $\prod_{j=1}^{m_x} S_{c_j}$, where $\{c_j\}$ is s.t. $\sum_{j=1}^{m_x} c_j = d$.[9] Therefore, we have

$$W_x(v) = \sum_{\varphi \in \text{Stab } x} \varphi \cdot v = \left( \sum_{\varphi \in \prod_{j=1}^{m_x} S_{c_j}} P_\varphi \right) v.$$

Note that $P_\varphi = \text{blkdiag}(P_1, P_2, \ldots, P_{m_x})$,[10] with each $P_j \in \mathbb{R}^{c_j \times c_j}$ being a permutation matrix, and the sum of all size $c_j$ permutation matrices is $(c_j - 1)! \mathbf{1}_{c_j \times c_j}$, where $\mathbf{1}$ denotes the all-ones matrix. Thus, by decomposing $W_x(v) \in \mathbb{R}^d$ into $\mathbb{R}^{c_1} \times \mathbb{R}^{c_2} \times \cdots \times \mathbb{R}^{c_{m_x}}$ we have

$$W_x(v) = \left( \frac{A_0}{c_1} \sum_{i=s_0+1}^{s_1} v_i \mathbf{1}_{c_1}, \quad \frac{A_0}{c_2} \sum_{i=s_1+1}^{s_2} v_i \mathbf{1}_{c_2}, \quad \ldots, \quad \frac{A_0}{c_{m_x}} \sum_{i=s_{m_x-1}+1}^{s_{m_x}} v_i \mathbf{1}_{c_{m_x}} \right),$$

where $A_0 = \prod_{j=1}^{m_x} c_j!$ and $s_j = \sum_{l \leq j} c_l$.

Let $e^{(j)} \in \mathbb{R}^d$ be such that $e_k^{(j)} = 1$ if $s_{j-1} < k \leq s_j$, and $e_k^{(j)} = 0$ otherwise. Then a sufficient condition for $W_x(v) = 0$ is that $\langle v, e^{(j)} \rangle = 0$ for all $j \in [m_x]$. Let $n_x \in N_x$ be an inward normal vector and fix $j \in [m_x]$. Since $x \pm \alpha_j e^{(j)} \in C_d$ for $\alpha_j = \min(x_{s_j} - x_{s_{j-1}}, x_{s_{j+1}} - x_{s_j}) > 0$, we conclude that $\langle n_x, \pm e^{(j)} \rangle \geq 0$ and hence $\langle n_x, e^{(j)} \rangle = 0$. Thus, $W_x(n_x) = 0$. $\qquad\square$

## C  Additional Results and Full Algorithm for Section 3.2

---

[9]For example, for $x \in C_5$ with $x_1 = x_2 < x_3 = x_4 < x_5$, the stabilizer is $S_2 \times S_2 \times S_1$.

[10]$\text{blkdiag}(P_1, P_2, \ldots, P_{m_x})$ is the block diagonal matrix $\begin{pmatrix} P_1 & 0 & \cdots & 0 \\ 0 & P_2 & \cdots & 0 \\ \vdots & \vdots & \ddots & \vdots \\ 0 & 0 & \cdots & P_{m_x} \end{pmatrix} \in \mathbb{R}^{d \times d}$.

Table 2: Semi-synthetic experiment: average-case test risk for the best stopping iteration, for $L = 2, W = 50$.

| Likelihood | Algorithm | boston | concrete | energy | kin8nm | naval | power plant | wine | yacht |
|---|---|---|---|---|---|---|---|---|---|
| Gaussian | MALA | 0.071 | 0.058 | 0.053 | 0.054 | 0.052 | 0.050 | 0.063 | 0.057 |
| | GD | 0.071 | 0.058 | 0.053 | 0.054 | 0.051 | 0.050 | 0.063 | 0.057 |
| | ULA | 0.070 | 0.058 | 0.053 | 0.054 | 0.051 | 0.050 | 0.063 | 0.057 |
| Laplacian | MALA | 0.194 | 0.172 | 0.170 | 0.167 | 0.161 | 0.160 | 0.186 | 0.167 |
| | GD | 0.194 | 0.173 | 0.170 | 0.168 | 0.161 | 0.160 | 0.187 | 0.168 |
| | ULA | 0.192 | 0.172 | 0.170 | 0.166 | 0.161 | 0.160 | 0.186 | 0.169 |

Table 3: Semi-synthetic experiment: average-case test risk for the best stopping iteration, for $L = 3, W = 50$.

| Likelihood | Algorithm | boston | concrete | energy | kin8nm | naval | power plant | wine | yacht |
|---|---|---|---|---|---|---|---|---|---|
| Gaussian | MALA | 0.069 | 0.059 | 0.055 | 0.056 | 0.051 | 0.052 | 0.068 | 0.053 |
| | GD | 0.070 | 0.059 | 0.056 | 0.056 | 0.051 | 0.051 | 0.069 | 0.054 |
| | ULA | 0.069 | 0.059 | 0.055 | 0.056 | 0.051 | 0.050 | 0.068 | 0.053 |
| Laplacian | MALA | 0.193 | 0.176 | 0.173 | 0.173 | 0.161 | 0.161 | 0.189 | 0.176 |
| | GD | 0.197 | 0.176 | 0.172 | 0.173 | 0.161 | 0.161 | 0.190 | 0.175 |
| | ULA | 0.194 | 0.176 | 0.172 | 0.172 | 0.161 | 0.161 | 0.190 | 0.174 |

---

**Algorithm 1** The algorithm for evaluating (6) with MALA

**Require:** A training set $\mathbf{X}_{\text{train}}$ and a test set $\mathbf{X}_{\text{test}}$; a BNN $f_{\text{BNN}}(\cdot; \theta)$ and a prior $p_\theta$; a likelihood $p(\cdot \mid \cdot)$.
**Ensure:** An approximation of (6).

1: **for** $i = 1, \ldots, 8$ **do**
2:      $\theta_*^{(i)} \sim p_\theta$             ▷ the groundtruth
3:      $\mathbf{Y}_{\text{train}}^{(i)} \sim p(\cdot \mid f_{\text{BNN}}(\mathbf{X}_{\text{train}}; \theta_*^{(i)}))$
4:      $\mathbf{Y}_{\text{test}}^{(i)} \sim p(\cdot \mid f_{\text{BNN}}(\mathbf{X}_{\text{test}}; \theta_*^{(i)}))$
5:      **for** $j = 1, \cdots, 50$ **do**
6:          $\theta_{\text{init}}^{(j)} \sim p_\theta$             ▷ the initial state
7:          $\theta_{\text{MALA}}^{(j)} \leftarrow \text{MALA}(\theta_{\text{init}}^{(j)}, \mathbf{X}_{\text{train}}, \mathbf{Y}_{\text{train}}^{(i)})$             ▷ the posterior sample
8:      **end for**
9:      **if** the likelihood is Gaussian **then**
10:          $\ell(\hat{y}, y) := (\hat{y} - y)^2$             ▷ the loss function derived from the likelihood
11:          $\hat{f}^{(i)}(x) := \frac{1}{50} \sum_{k=1}^{50} f_{\text{BNN}}(x; \theta_{\text{MALA}}^{(k)})$             ▷ the predictive function
12:      **else if** the likelihood is Laplacian **then**
13:          $\ell(\hat{y}, y) := |\hat{y} - y|$
14:          $\hat{f}^{(i)}(x) := \text{ median of } \{y^{(k)} : y^{(k)} \sim p(\cdot \mid f_{\text{BNN}}(x; \theta_{\text{MALA}}^{(k)})), k = 1, \ldots, 50\}$
15:      **end if**
16:      $L^{(i)} \leftarrow \frac{1}{|\mathbf{X}_{\text{test}}|} \sum_{(x,y) \in (\mathbf{X}_{\text{test}}, \mathbf{Y}_{\text{test}}^{(i)})} \ell(\hat{f}^{(i)}(x), y)$
17: **end for**
18: $L \leftarrow \frac{1}{8} \sum_{i=1}^{8} L^{(i)}$             ▷ the approximated average-case risk (6)

---

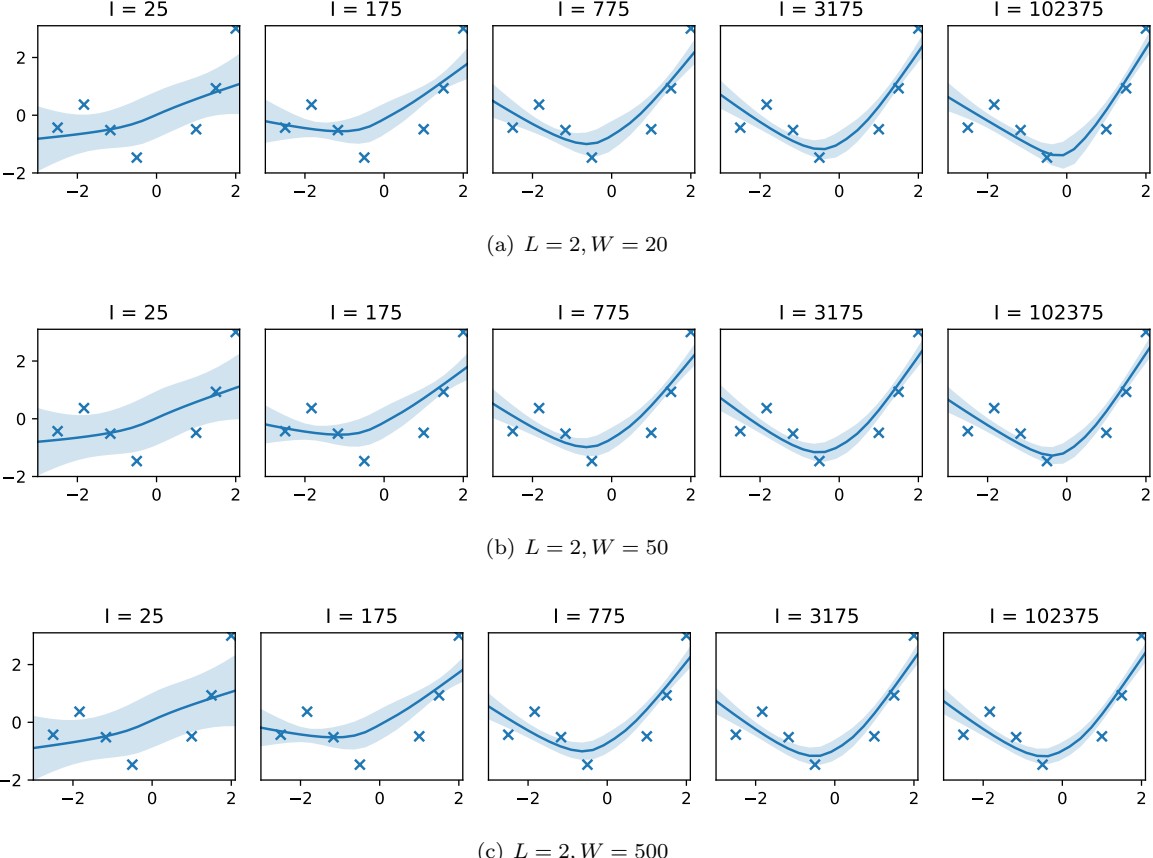

(a) $L = 2, W = 20$

(b) $L = 2, W = 50$

(c) $L = 2, W = 500$

Figure 4: Additional visualizations in the setting of Fig. 1.

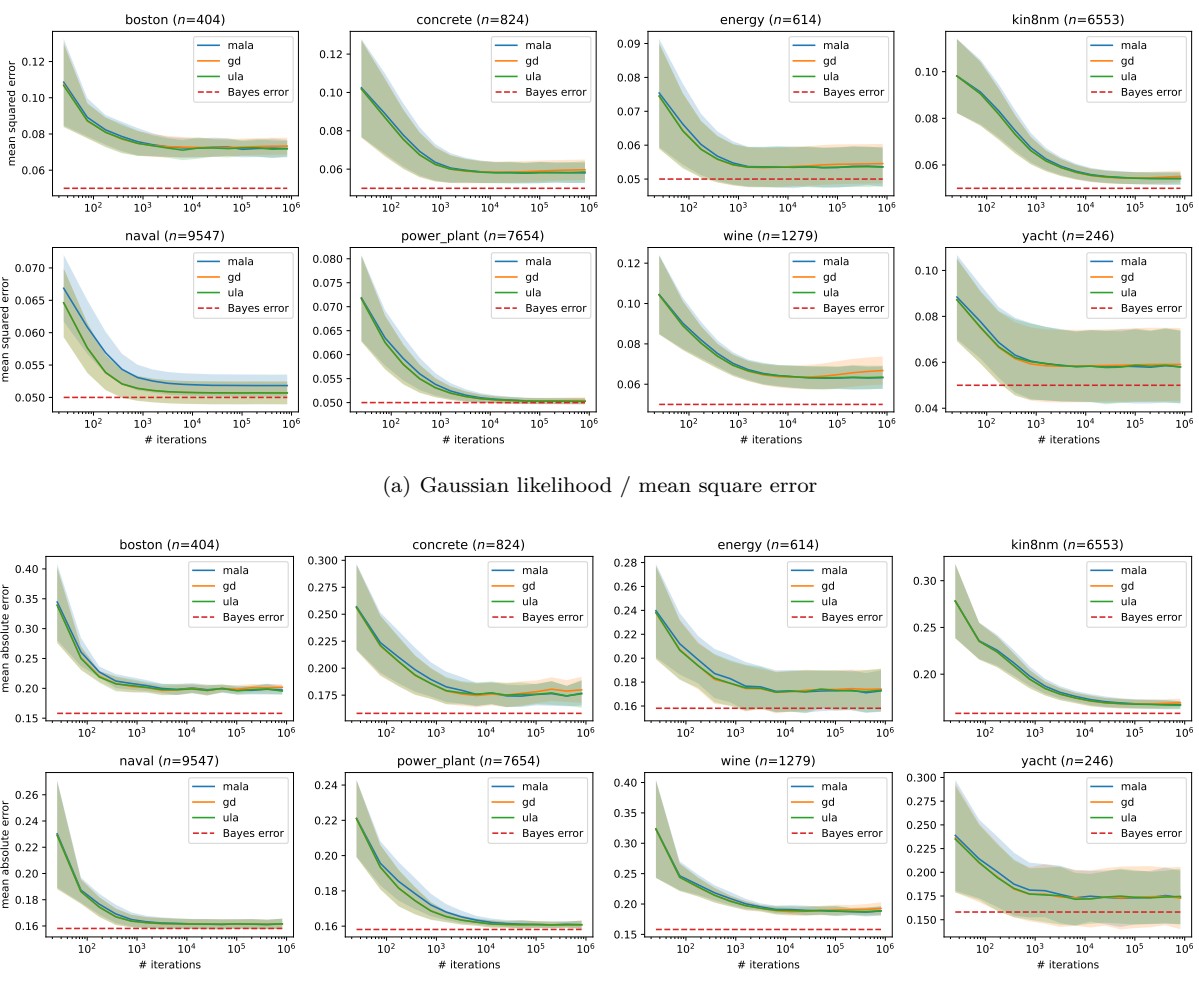

(a) Gaussian likelihood / mean square error

(b) Laplace likelihood / mean absolute error

Figure 5: Semi-synthetic experiment: estimated loss (6) under different likelihoods, for $L = 2, W = 50$.

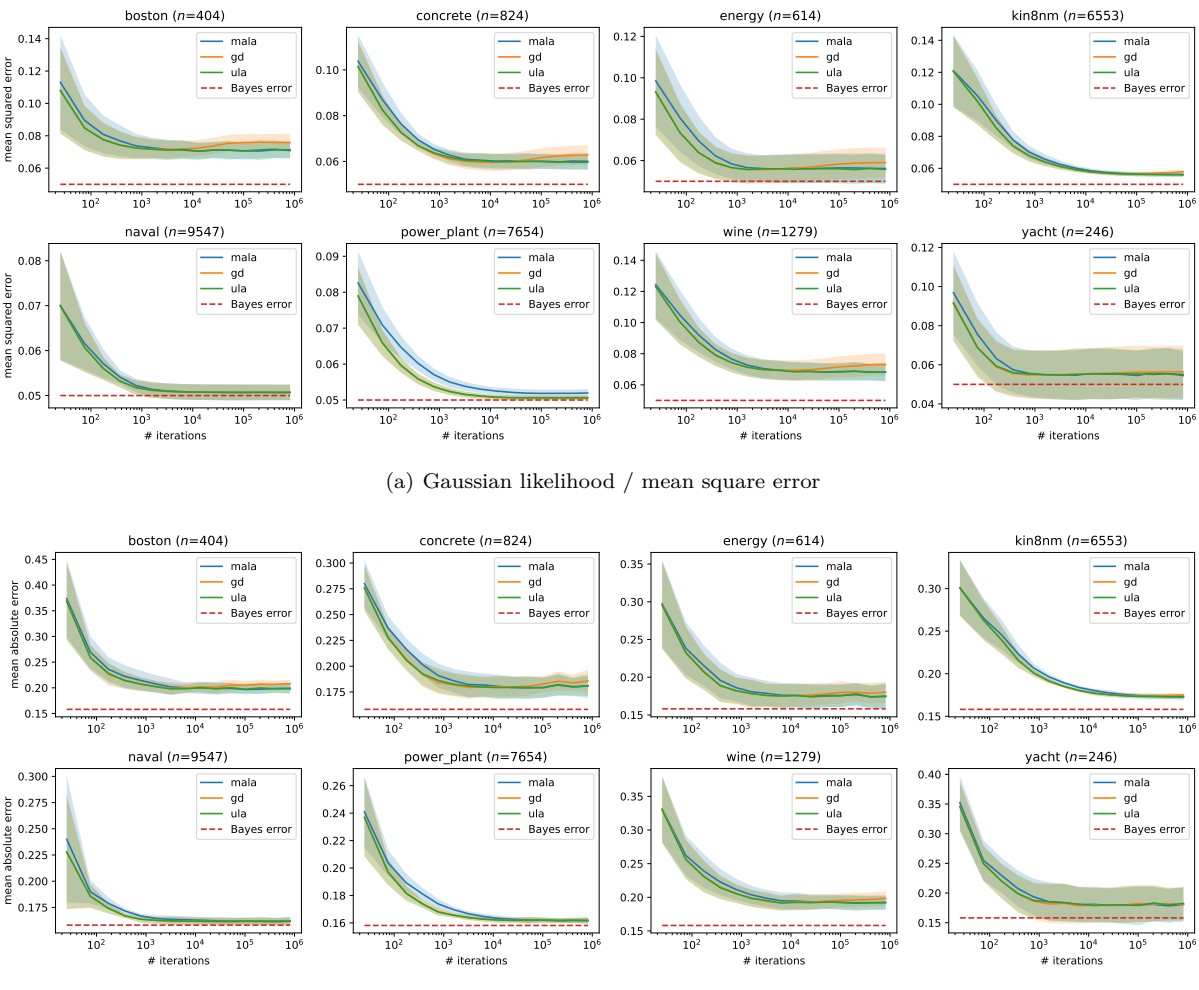

(a) Gaussian likelihood / mean square error

(b) Laplace likelihood / mean absolute error

Figure 6: Semi-synthetic experiment: estimated loss (6) under different likelihoods, for $L = 3, W = 50$.

