# OpenReview forum: "On Equivalences between Weight and Function-Space Langevin Dynamics"
_TMLR — Rejected by TMLR_

### Review · Reviewer_seUs · 2023-05-22

**Summary Of Contributions:**

The paper shows that weight and function space inference of BNNs are in equivalent in some sense.

**Audience:**

Yes

**Broader Impact Concerns:**

No issues

**Claims And Evidence:**

No

**Requested Changes:**

This paper shows some theoretical results between weight and function space langevin dynamics. The main results is a statement that they are “equivalent”. This is a bold statement, and does not come with many disclaimers. I feel that the paper’s conclusions are too grand and not precise enough, and not supported by enough evidence. The paper is a bit vague in what equivalency means, and the exact contributions of this paper are not clear. I feel the paper is a bit roundabout: some results are shown but there is no clear statement of what are the theoretical contributions. In the end I feel that this paper mostly states that parameter posteriors can be trivially turned into function posteriors, which feels quite an obvious result.

There is also some remarks about overparameterisation, which are interesting but felt incomplete.

I wonder how significant is the analysis of equivalences between WLD and FLD. Surely the main interest is in the sample-efficiency, and in convergence properties. WLD should be less sample-efficient due to the symmetries, permutations and invariances in the weight space. I wonder how this paper’s claims then relate to these two aspects.

The experiments are small scale, and it was not clear what claims they are studying or demonstrating. The experiments show that MALA can sample small networks. Ok, but large-scale LD results are common-place in literature, so what is the added insight from this paper?

The authors need to clarify what are the precise claims of the paper, and how do the empirical analysis study or verify these claims. The paper is also mathematically quite dense, but frustratingly sloppy in notation, and author clarifications are needed.

- I don’t get the A notation. It maps the parameters space to F, which is a subset of R^n. What does this mean? Do we assume that outputs $y$ are real? Do we assume the function space contracts to training data only? Aren’t functions infinite?
- I also don’t understand what A_# \pi means. “A” should map parameters to functions, but now it maps densities into densities. How can you apply R^d → R^n operator to R^d → R^1 function? It seems some notation overloading is happening, please define this.
- How does the push-forward handle change of volume? It would be good to give a precise definition of the push-forward, since $A_# \pi_\theta$ is right now undefined.
- I’m confused by the size of F: at first it’s defined as subset of R^n, but later in FLD its defined as less than d. So is F then up to R^d?
- What is “riemannian manifold structure”? Please define. What is $k$? Why do we choose low-dimensional stuff here. This seems abrupt and even harmful: surely we want to cover the function space as-is, and not some contraction of it. The FLD defines tilde-f_t as one coordinate of R^n size f, yet this “one coordinate” has size k.
- What is size of G? What are assumptions about G? What is g? What does the strange $g^{ij}_{ij}$ notation mean? What is a “coordinate matrix”? What is coordinate (is it one datapoint..)? What is our metric? What is i? What is j? What is f(f)? What is partial_j? What is “corresponding Riemann measure”?
- The math overall needs more precision: please define all symbols and terms precisely, and also give intuitive conceptual explanations and motivations. Right now parts of the story and technicalities are missing.
- It is not “intuitive” to me that WLD equals FLD. The paper cites Girolami2011, but this does not contain any discussion about function-space sampling or pushforwards. If the WLD/FLD connection is known, can you add a citation; and if not, can you derive it yourself?
- What is $\cal{X}$? What is $|\cal{X}|$? How can you use $\cal{X}$ inside the definition of $\cal{X}$, isn’t this circular? To me a natural definition for input space would be $R^D$ or some subset of it. Here instead we define it as a finite set, which feels odd. Does this then refer to $n$ datapoints instead? Is this based on some kind of Dirac measure? Are you talking about finite dataset or the input domain here? I’m pretty lost since I can’t understand what your math means..
- “above vector representation”.. errr where? I don’t see any vector representation. What does “coordinate” of F mean? I hope F still means a subset of R^n, but at this point I’m unsure since it does not seem to align with the rest of the math. It might be a good idea to remind the reader of definition of $\cal{F}$.
- What is “coordinate matrix” of A? What is its size?
- What does it mean that “in this coordinate the metric has a coordinate AA”. I assume that coordinate refers to a single axis or dimension, but suddenly coordinate also means a matrix. I’m lost and don’t really understand the high-level story anymore.
- Please define the symbols and their domains and their sizes in the ex 2.1. equations.
- Ex 2.1. (ii) is quite technical, and there is no proof or derivation or citation for this result. Can you add it? Where does this result come from? “ran” and “ker” are undefined.
- Can you define $(\psi_1 \psi_2)$ and $\psi \cdot p$. Both seem unintuitive, and are undefined.
- $A|_p u$ is undefined. $T_{Ap} F$ is undefined. $T_p(H \cdot p)^{I}$ is undefined. Please also give conceptual explanations of every symbol in these (eg. what does p,u,T,.. represent?)
- What is a discrete group, please define. This sounds odd: surely parameters have continuous invariances?
- The prop 2.1. is strangely written. The first sentence assumes some terms with no connection to posteriors or parameters. Then suddenly the second sentence talks about prior and WLD and FLD, even though there is no connection between them and the A and H in the context of prop 2.1. Perhaps a connecting phrase is missing. I don’t really understand what this proposition tries to say.
- WHat does “equivalence” of WLD and FLD mean? This is vague wording.
- I see no relevance of example 2.2. due to its artificiality. No ML function looks like this (mapping parameters to functions by taking decimals). What’s then the point?
- The example 2.2. italics continues for a long time. Is all of this intended to be part of the example? Does the v() equation have a plus? It looks like it should be equals.
- I don’t agree with example 2.3.: one can produce same function values without permutations. For instance, if we have a dataset of one datapoint, the function space (which is just real line) can be surjective. This of course depends on how the function space is defined (this is a bit loose). I think the example is assuming that function space is infinite, while earlier definition assumed it to be $R^n$.
- It’s unclear what does $\theta_t$ refer to in ex 2.3. I assume its weight LD, but not sure. It’s unclear what does $\tilde{p}(\theta)$ mean: how can you evaluate parameters in a function density? The dimensions won’t match. I’m now confused what tilde p is trying to be.
- What is “function space measure induced by WLD”? How do you compute or estimate it? What does “credible interval” mean? Please specify. What does “marginal” distribution mean here? What do you marginalise? What is “marginal distribution q induced by WLD”? Please define. How do you obtain ground truth?
- I don’t understand what eq 1 means. Does this have something to do with KSD? I general I couldn’t follow the KSD stuff, or what really happens. I get that you use a sampler to get the parameter posterior, but everything after that is unclear.
- Earlier H was a group, now its width. Please avoid this.
- The paper talks about LD, but experiments are using MALA. Is this ok? How can you say that the results from MALA transfer to LD? Why not use LD in the experiments if the paper is about LD..? MALA is also undefined.
- It’s a bit unclear what you are demonstraing in figure 1. So it seems that we get better fits with more samples. Ok sure, but what’s the point? What research question or claim are you studying here? I don’t see a connection to the contributions of this paper.
- Fig2: I don’t know what “LD predictive distribution” or “approximate functoin space posterior” means. Can you define them?
- Fig2: It seems that the MALA converges wrt something. What is the point of this figure, and what research question are you demonstrating or studying here? I can’t follow.
- In 3.2. I can’t follow what you do. It seems that you sample many $\theta_0$, which means that you sample a lot of ground truth functions. Yet do you only do MALA once..? I can’t follow what’s going on. What does 8 $\theta_0$’s mean? Are these the ground truth functions, or draws from the prior? I think the text is referring that we sample bunch of stuff from the prior, but surely this does not fit the data in any way. WHat is then the point?
- It would be very helpful to have algorithm boxes to describe the experiments.
- I also wonder about the mean arguments: surely in a multimodal posterior you should not average since you can end up between peaks.
- I’m having in general trouble seeing the research question studied in 3.2. Suddenly we have ensembles entering the discussion here, which complicates the discussion and makes it difficult to see what hypothesis you are studying here. The section concludes by saying that MALA is a good method for BNNs. Ok, but isn’t this already known? For instance, in Izmailov’2021 they used LD to sample large neural networks and showed good performance. What is the novelty of your demonstration?

**Strengths And Weaknesses:**

S: The research topic is interesting and timely.

S: The results have theoretical significance

W: The paper is sloppily written

W: It's unclear how significant the results are.

W: The experiments are not convincing or particularly clear.

---

> ### Author Response · Authors · 2023-06-19
> **Response (Part 1)**
>
> Thank you for your valuable feedbacks and suggestions. We apologize for the inconvenience of reading this article. We have thoroughly revised the submission to improve readability, and have made the following changes based on your suggestions:
>
> - We have added references to the mathematical notions employed in the main text and a background section (Appendix A) to provide an introduction of relevant concepts. See our responses to your Q7, Q8, Q10-Q15 and Q20 for details.
> - We have made the meaning of "equivalence" precise, and provided the detailed proof of Example 2.1. See our responses to your Q1, Q16 and Q17.
> - We have revised Section 1, Section 3 and Section 4 to clarify on the purpose of the experiments. We have also added an algorithm box to describe the experiment in Section 3.2. See our responses to your Q21, Q23-Q26 and Q28.
>
> Before addressing your questions, we want to clarify on some important misunderstandings about our notations:
>
> - $\\mathcal X$ is the input space and $n$ is the size of training set. The cardinality of $\\mathcal X$ is denoted by $|\\mathcal X|$ and it differs from $n$. See our response to your Q6 for details.
> - $\\mathcal F$ is the function space and it is never a subset of $\\mathbb R^n$. See our response to your  Q6 and Q10 for details.
>
> We will now address your questions, which we grouped into 6 sections below.
>
> ### Main Claims and Contributions
>
> > **[Q1a.]** The paper is a bit vague in what equivalency means, and the exact contributions of this paper are not clear.
>
> 1. The revision has clarified on **the meaning of equivalence**: let the trajectory $\\{\\mathcal{A}\\theta\_t\\}$ be derived from (WLD), and $\\{f\_t\\}$ be from (FLD) or its respective generalization. Then the "equivalence" between (WLD) and (FLD) means that for any $k\\in\\mathbb{N}$ and $\\{t\_i\\}\_{i=1}^k\\subset [0, \\infty)$, $\\{\\mathcal{A}\\theta\_{t\_i}\\}\_{i=1}^k$ and $\\{f\_{t\_i}\\}\_{i=1}^k$ equal in distribution.
> 2. We are interested in this notion of equivalence primarily because when it holds, the simulation of (WLD) can be viewed as a "function-space inference" algorithm, in the sense that the distribution of $\\mathcal{A} \\theta\_t$ will converge to the function-space posterior at a rate which is not hindered by the overparameterization of the model: the distribution is equivalent to that of $f\_t$ from (FLD), which does not have to be defined using an overparameterized model.
> 3. Even though such an equivalence does not necessarily imply a rapid convergence to the function-space posterior, the fact that the simple algorithm[^1] of Langevin dynamics (LD) fulfills the criterion of "function-space inference" is worth noting, in light of the apparent benefit of such an idea, and the limitation of many existing works that design dedicated inference schemes specifically to fulfill this criterion. **This is a main message of this work**.
> 4. However, there are also settings where **the equivalence allows us to gain concrete insights** into the behavior of (WLD), such as on overparameterized linear models which formally covers DNNs in the "kernel regime". We have reorganized the relevant discussions into a new Section 2.2 in the revision.
>
> [^1]: Note that technically, our analysis did not cover the discretization errors which are unavoidable in the actual algorithmic implementations. However, our experiments provided some (indirect) evidence on the efficacy of the practical algorithms.
>
> > **[Q1b.]** In the end I feel that this paper mostly states that parameter posteriors can be trivially turned into function posteriors, which feels quite an obvious result.
>
> Given the above explanation, it should be clear now that this is not the case.
>
> > **[Q2.]** There is also some remarks about overparameterisation, which are interesting but felt incomplete.
>
> We have updated the discussion below Example 2.1, and added a new Section 2.2 in the revision, where we discussed in more detail how the equivalence result for overparameterized linear models may enable us to understand the behavior of WLD on BNNs.
>
> > **[Q3.]** WLD should be less sample-efficient due to the symmetries [...] in the weight space. I wonder how this paper’s claims then relate to these two aspects.
>
> The equivalence between (the function-space pushforward of) WLD and FLD implies that even though WLD is inefficient in approximating the weight-space posterior, **the inefficiency arising from overparameterization is irrelevant** in predictive modeling, because we only care about the pushforward measure in function space (i.e., the distribution of $\\mathcal A\\theta\_t$), and WLD is equivalent to FLD in this aspect.
>
> (As noted in our last response, this does not always imply WLD is a fully efficient algorithm, becuase FLD isn't, but there are concrete models for which we can understand this issue.)

---

> > ### Author Response · Authors · 2023-06-19
> > **Response (Part 2)**
> >
> >
> > > **[Q4a.]** it was not clear what claims [the experiments] are studying or demonstrating.
> >
> > We have revised Section 1, Section 3 (the beginning and Section 3.1) and Section 4 to clarify on the purpose of the experiments:
> >
> > * The experiments aim at evaluating the convergence of the weight-space Langevin-type algorithms in the function-space. They do not provide direct evidence for the equivalence, but rather for a *possible consequence* of it: when the model is correctly specified and the metric in (FLD) is not known to have an adverse effect, the function-space distributions may have good convergence properties (see Sec 2.2).
> > * These experiments also share a high-level goal with our analysis, which is to understand the behavior of Langevin-type algorithms on (overparameterized) BNNs. In this aspect, they also complement past works with a similar goal, by analyzing practical models (instead of infinitely wide models as in past theoretical works) and by eliminating the influence of model misspecification (which appears in the past empirical works cited in paper).
> >
> >
> > > **[Q4b.]** The experiments are small scale. [...] large-scale LD results are common-place in literature, so what is the added insight from this paper?
> >
> > * Our experiments analyze the convergence of (the pushforward of) Langevin-type algorithms to the function-space posterior, **in controlled settings that eliminated the influence of model misspecification.** To our knowledge, past results on LD for large-scale BNNs (Izmailov et al, 2021; Wenzel et al, 2020) mostly focus on predictive performance on real-world datasets, in which we cannot rule out the influence of model misspecification. This is especially relevant because prior misspecification may be especially relevant in their image classification setup (Fortuin et al, 2021).
> > * The scale of the experiment in Section 3.1 is limited by the hardness of estimating the KSD, which involves score estimation for the function-space prior. This is not encountered by past studies of LD for BNNs, which did not analyze the convergence in (function-space) KSD.
> >
> > We have updated Section 3 to provide better visibility to the above points.
> >
> > > **[Q5.]** The authors need to clarify what are the precise claims of the paper, and how do the empirical analysis study or verify these claims.
> >
> > Please refer to our response to Q1a for the precise theoretical claims, and to Q4a on the goal of the empirical analysis. We have updated the text to clarify on these points, as referenced in our response.

---

> > > ### Author Response · Authors · 2023-06-19
> > > **Response (Part 3)**
> > >
> > >
> > > ### Questions about manifolds / function spaces
> > >
> > > > **[Q6a.]**  (the third paragraph of Sec. 1) I don’t get the A notation. It maps the parameters space to F, which is a subset of R^n. What does this mean? Do we assume that outputs are real? Do we assume the function space contracts to training data only?
> > > >
> > > > **[Q6b.]**  What is $\\mathcal X$? What is $|\\mathcal X|$? To me a natural definition for input space would be $R^D$ or some subset of it. Here instead we define it as a finite set, which feels odd. Does this then refer to datapoints instead? Are you talking about finite dataset or the input domain here?
> > > >
> > > > **[Q6c.]**  (around equation (FLD)) I’m confused by the size of F: at first it’s defined as subset of R^n, but later in FLD its defined as less than d. So is F then up to R^d?
> > >
> > > * **We have changed a possibly misleading notation** in the definition of $\\mathcal F$: we now state that $\\mathcal F\\subset\\mathbb R^{\\mathcal X} := \\{ \\text{real-valued functions } f: \\mathcal X \\to \\mathbb R \\}$, instead of $\\mathbb R^{|\\mathcal{X}|}$. Our previous notation only makes sense when $\\mathcal X$ has finite cardinality; we apologize for the confusion. We assume the outputs are real ($y\_i\\in\\mathbb{R}$).
> > > * $\\mathcal X$ is the input space which is a superset of the training inputs $\\{x\_1\\}\_{i=1}^n$. $|\\mathcal{X}|$ denotes its cardinality, which can be infinity (such as in Ex 2.2 or 2.3) or finite (Ex 2.1). $n$ is the size of the training dataset. We do not assume $\\mathcal{X}$ contracts to the training inputs, and thus $n\\ne |\\mathcal X|$ in general.
> > > * Please note that $\\mathcal X$ never refers to the (training) "data points", and **$\\mathcal{F}$ is never a subset of $\\mathbb{R}^n$**.
> > > The relevant sentence around (FLD) means (to assume that) $\\mathcal{F}$ is a manifold with dimensionality up to $d$.
> > >
> > > > **[Q6d.]**  (around equation (FLD)) What is k? Why do we choose low-dimensional stuff here.
> > >
> > > $k$ is the dimensionality of Riemannian manifold $\\mathcal F$. We are not "choosing" to assume $k\\le d$; it follows immediately by the fact that $\\mathcal{F}$ is the image of $\\mathcal{A}$ and the input domain of $\\mathcal{A}$ is a subset of $\\mathbb{R}^d$. In other words, we are not switching to a "contraction" of the function space here.
> > >
> > > > **[Q6e.]** I think [example 2.3] is assuming that function space is infinite, while earlier definition assumed it to be $\\mathbb{R}^n$.
> > >
> > > As noted in our response to Q6a-Q6c the function space is never $\\mathbb{R}^n$. We take your question as asking if $\\mathcal{F}$ is infinite-dimensional in this example. Here $\\mathcal{F}$ is finite-dimensional, because we have constructed a coordinate map using the cone $C\_n\\subset \\mathbb{R}^d$. (Alternatively we can observe that its dimensionality is at most $d$, by the argument in our response to Q6d.)
> > >
> > > > **[Q7a.]** I also don’t understand what A_# \\pi means.
> > > >
> > > > **[Q7b.]** How does the push-forward handle change of volume?
> > >
> > > * $\\mathcal A\_\\#$ refers to the pushforward (of a measure). In the revision we have referenced a definition (Villani, 2009, p.11).
> > > * It is a often-used result that the pushforward does not need to handle the change of volume. See for example, Theorem 3.6.1 in *Measure Theory, Volume I* by Borgachev (2007).

---

> > > > ### Author Response · Authors · 2023-06-19
> > > > **Response (Part 4)**
> > > >
> > > >
> > > > > **[Q8a.]**  What is “riemannian manifold structure”? Please define.
> > > > >
> > > > > **[Q8b.]**  What is size of G? What are assumptions about G? What is g? What is a “coordinate matrix”? What does the strange $g_{ij}$, $g^{ij}$ notation mean? What is coordinate (is it one datapoint..)? What is our metric? What is i? What is j? What is $f(\\tilde f)$? What is partial_j? What is “corresponding Riemann measure”?
> > > >
> > > > The notions and notations about differential/Riemannian geometry are taken from Lee (2018). The revision now includes references to the definitions in the text, and an introductory section on Riemannian manifolds in Appendix B.2.  Below we also provide brief answers to your questions.
> > > >
> > > > - The Riemannian structure on a manifold is a generalization of the inner product in Euclidean spaces.
> > > > - A $k$ dimensional manifold locally looks like the Euclidean space $\\mathbb R^k$, and we can use points in $\\mathbb R^k$ to represent points on manifold. This representation is a _coordinate_, and it is not a datapoint.
> > > > - Given a coordinate:
> > > >   - $i$ and $j$ are the $i$-th and the $j$-th components of the coordinate.
> > > >   - The inner product at a point $p$ on manifold can be represented as an inner product in $\\mathbb R^k$ defined by the matrix $G \\in \\mathbb R^{k \\times k}$, i.e., the inner product of $u, v \\in \\mathbb R^k$ is $u^\\top G v$. We assume $G$ smoothly depends on $p$.
> > > >   - $g\_{ij}$ and $g^{ij}$ are the $(i, j)$-th elements of $G$ and $G^{-1}$, respectively. These notations are not strange; they are the standard conventions in Riemannian geometry (see Lee (2018) and Do Carmo (1992)).
> > > >   - $\\partial\_j$ represents taking partial derivative w.r.t. the $j$-th component.
> > > > - $f(\\tilde f)$ is the function in $\\mathcal F$ corresponding to the coordinate $\\tilde f$.
> > > > - The Riemann measure of a set on $\\mathcal F$ can be understood as the volume of this set. It is the generalization of the volume (Lebesgue measure) in Euclidean spaces.
> > > >
> > > > > **[Q9.]** (a question around the equation (FLD)) It is not “intuitive” to me that WLD equals FLD. The paper cites Girolami2011, but this does not contain any discussion about function-space sampling or pushforwards. If the WLD/FLD connection is known, can you add a citation; and if not, can you derive it yourself?
> > > >
> > > > The sentence that included the word "intuitive" stated that it is intuitive that we can sample from $\\pi\_{f|\\mathcal{D}}$ using Riemannian LD. The citation is about the work on Riemannian LD. The sentence does not make any claims on the equivalence, which is derived in the subsequent examples and propositions.
> > > >
> > > > ### Questions about Example 2.1
> > > >
> > > > > **[Q10a.]** [What does vector representation means.] What does “coordinate” of F mean?
> > > > > **[Q10b.]**  I hope F still means a subset of $\\mathbb R^n$.
> > > >
> > > >
> > > > - When the input space $\\mathcal X = \\{ \\mathtt x\_1, \\mathtt x\_2, ..., \\mathtt x\_K \\}$[^31] is finite, a function $f: \\mathcal X \\to \\mathbb R$ can be uniquely represented as a vector $(f(\\mathtt x\_1), f(\\mathtt x\_2), ..., f(\\mathtt x\_K)) \\in \\mathbb R^{K}$, which is the "vector representation".
> > > > - As we have clarified in Q6, $\\mathcal F \\subset \\mathbb R^{\\mathcal X}$ is still the function space. In this example, the input space is finite and $\\mathbb R^{\\mathcal X}$ is identified with $\\mathbb R^K$ via the vector representation.
> > > > - In the bijective case, $\\mathcal F = \\mathbb R^{\\mathcal X} = \\mathbb R^K$, so we use a vector $v \\in \\mathbb R^K$ as the _coordinate_ of the corresponding function in $\\mathcal F$.
> > > > - We have never stated $\\mathcal{F}\\subset\\mathbb R^n$. $n$ is the number of training samples, so this cannot be true.
> > > >
> > > > [^31]: please also note that $\\mathtt{x}\_1$ (element of input space) and $x\_1$ (training input) are different symbols.
> > > >
> > > > >  **[Q11.]** What is “coordinate matrix” of A? What is its size? What does it mean that “in this coordinate the metric has a coordinate $(AA^\\top)^{-1}$”.
> > > >
> > > >
> > > > - Metric and coordinate are notions in Riemannian geometry; see our response to your Q8.
> > > > - The phrase "coordinate matrix" in $A$ reflects another usage, for linear maps. To avoid possible confusion, we have now removed this occurrence. It now states that (in that example) $\\mathcal A$ is a linear map between $\\mathbb{R}^d$ and $\\mathcal{F}\\subset\\mathbb{R}^K$, so we can identify it with a matrix $A \\in \\mathbb R^{K \\times d}$. More specifically, for any $\\theta\\in\\mathbb{R}^d$, $A\\theta = (f(\\mathtt x\_1;\\theta),\\ldots,f(\\mathtt x\_K;\\theta))\\in\\mathbb{R}^K$.
> > > > - We also changed the sentence "in this coordinate the metric has a coordinate $(AA^\\top)^{-1}$" to "in this coordinate the metric matrix $G$ is $(AA^\\top)^{-1}$".

---

> > > > > ### Author Response · Authors · 2023-06-19
> > > > > **Response (Part 5)**
> > > > >
> > > > >
> > > > > > **[Q12.]** Please define the symbols and their domains and their sizes
> > > > >
> > > > > We have revised that example and added the definitions in our preceding answers.
> > > > >
> > > > > > **[Q13.]** Ex 2.1. (ii) is quite technical, and there is no proof or derivation or citation for this result. [...] “ran” and “ker” are undefined.
> > > > >
> > > > > The proof idea was given in the example. We have now added a full proof in Appendix B.1. "Ran" and "Ker" denote the range and the kernel of a linear map.
> > > > >
> > > > > ### Questions about Proposition 2.1
> > > > >
> > > > > > **[Q14a.]** Can you define $(\\psi_1\\psi_2)$ and $\\psi \\cdot p$.
> > > > >
> > > > > Thank you for point them out. They are notations from group theory. We have referenced our use of these notations in the revision, and added a background subsection in Appendix A. We also provide the definitions below:
> > > > >
> > > > > - The notation $\\psi \\cdot p$ represents the group action. Recall that a group action is a map $\\Gamma: H \\times \\mathbb R^d \\to \\mathbb R^d$, we use $\\psi \\cdot p$ to denote $\\Gamma(\\psi, p)$ for $\\psi \\in H, p \\in \\mathbb R^d$.
> > > > > - The notation $\\psi\_1 \\psi\_2$ is an abbreviation of the operation $\\Psi(\\psi\_1, \\psi\_2)$ defined below.
> > > > > A group $H$ is a set equipped with an operation (someone calls this multiplication) $\\Psi: H \\times H \\to H$. It satisfies the following properties:
> > > > >   - there is a unit element in $H$ (denoted as $e$) such that $\\Psi(e, \\varphi) = \\Psi(\\varphi, e) = \\varphi$ for every $\\varphi \\in H$.
> > > > >   - every element $\\varphi \\in H$ has an inverse (denoted as $\\varphi^{-1} \\in H$) such that $\\Psi(\\varphi, \\varphi^{-1}) = \\Psi(\\varphi^{-1}, \\varphi) = e$.
> > > > >   - the operation $\\Psi$ is associative, i.e., for $\\varphi\_1, \\varphi\_2, \\varphi\_3 \\in H$, it holds $\\Psi(\\varphi\_1, \\Psi(\\varphi\_2, \\varphi\_3)) = \\Psi(\\Psi(\\varphi\_1, \\varphi\_2), \\varphi\_3)$.
> > > > >
> > > > >
> > > > >
> > > > > > **[Q14b.]** $dA|\_p u$, $T\_{\\mathcal Ap} \\mathcal F$, $T\_p(H \\cdot p)^{\\perp}$ is undefined.
> > > > >
> > > > > Thank you for point them out. They are notations from differential geometry. We have referenced the definitions  in the revision, and added a explanatory subsection in Appendix A. We also provide the definitions below:
> > > > >
> > > > > - $T\_{\\mathcal A p} \\mathcal F$ is the tangent space (tangent plane) of the manifold $\\mathcal F$ at $\\mathcal Ap$.
> > > > > - The orbit $H\\cdot p$ is a submanifold of $\\mathbb R^d$. Its tangent space at $p$ is $T\_p(H \\cdot p)$ and can be viewed as a subspace of $T\_p \\mathbb R^d$. Given the Riemannian metric, the orthogonal complement $T\_p(H \\cdot p)^\\perp$ can be defined as $\\{ v \\in T\_p \\mathbb R^d : \\langle v, w \\rangle = 0, \\forall w \\in T\_p(H \\cdot p) \\}$.
> > > > > - $\\mathrm d \\mathcal A|\_p$ is the differential of $\\mathcal A$ at $p$. It is a linear map $T\_p \\mathbb R^d \\to T\_{\\mathcal Ap} \\mathcal F$. Therefore, $\\mathrm d \\mathcal A|\_p u$ is a tangent vector in the tangent space of $\\mathcal F$ at $\\mathcal Ap$. We changed this notation to $(\\mathrm d \\mathcal A|\_p)(u)$ for clarity.
> > > > >
> > > > > > **[Q15.]** What is a discrete group, please define. This sounds odd: surely parameters have continuous invariances?
> > > > >
> > > > > - A Lie group $H$ is said to be a discrete group (see e.g., Hall, 2013, p.28) if for each $\\varphi \\in H$, there exists a neighborhood $U$ of $\\varphi$ such that no points other than $\\varphi$ are contained in this $U$. Discrete group are useful for characterizing "global overparametrization" such as those caused by the permutation invariance in Example 2.3.
> > > > > - Continuous invariances are covered separately in Example 2.1. We have added a introductory paragraph before the heading of Section 2.1 to explain this point.
> > > > >
> > > > > > **[Q16.]** [in Proposition 2.1] The first sentence assumes some terms with no connection to posteriors or parameters. Then suddenly the second sentence talks about prior and WLD and FLD.
> > > > >
> > > > > - The assumption in the first sentence describes the invariance of the parameterization (characterized by $H$). It is connected to parameters through the parameterization map $\\mathcal A$.
> > > > > - The conclusion in the second sentence is the relationship of $\\mathcal A\\theta\_t$ and $f\_t$, which needs the assumption on $\\mathcal A$.
> > > > >
> > > > > > **[Q17.]** What does “equivalence” of WLD and FLD mean?
> > > > >
> > > > > Please see our response to your Q1a.

---

> > > > > > ### Author Response · Authors · 2023-06-19
> > > > > > **Response (Part 6)**
> > > > > >
> > > > > >
> > > > > > ### Questions about Examples 2.2 and 2.3
> > > > > >
> > > > > > > **[Q18a.]** I see no relevance of example 2.2 No ML function looks like this
> > > > > >
> > > > > > * We have updated the discussion to better emphasize that Example 2.2 is a concrete, albeit artificial example in which the equivalence allows us to deduce the fast convergence of $\\mathcal{A}\\theta_t$ to the function-space posterior.
> > > > > > * As stated in the text, it also highlights a different behavior of VI and MCMC methods on overparameterized models.
> > > > > > * The revision also better emphasizes that the relevant ML examples are Example 2.1 and 2.3.
> > > > > >
> > > > > > > **[Q18b.]** Is all of [the italics in Example 2.2] intended to be part of the example? Does the v() equation have a plus? It looks like it should be equals.
> > > > > >
> > > > > > Yes. No; please note that the denominator of the first summand uses $k\_h(\\theta^{(j)},\\theta^{(k)})$ while the second uses $k\_h(\\theta,\\theta^{(k)})$. We have now colored the first term for visibility.
> > > > > >
> > > > > > > **[Q20.]** It’s unclear what does $\\theta\_t$ refer to in ex 2.3.[...] It’s unclear what does $\\tilde p(\\theta)$ mean.
> > > > > >
> > > > > > - You are right $\\{ \\theta\_t \\}$ is the trajectory of WLD. We have reorganized Section 3.1 to show that this notation applies "globally".
> > > > > > - In this example, $\\mathcal A$ is bijective restricted to the cone $C\_n$, and $\\theta \\in C\_n$ is the coordinate of the function $\\mathcal A\\theta$.  Under the above coordinate, $\\tilde p(\\theta)$ is regarded as the evaluation of $\\tilde p$ at $\\mathcal A \\theta$.

---

> > > > > > > ### Author Response · Authors · 2023-06-19
> > > > > > > **Response (Part 7)**
> > > > > > >
> > > > > > >
> > > > > > > ### Questions about experiments
> > > > > > > > **[Q21a.]** What is “function space measure induced by WLD”? How do you compute or estimate it? What does “credible interval” mean? What is “marginal distribution q induced by WLD”? How do you obtain ground truth?
> > > > > > >
> > > > > > > - The function space measure induced by WLD is the distribution of $\\mathcal A \\theta\_t$. We do not directly compute it. Instead, we estimate its finite-dimensional marginal distribution $\\mathcal (\\mathcal A\\theta\_t)(\\mathbf X\_e)$, which is referred to as "marginal distribution $q$ induced by WLD".
> > > > > > > - The 80% pointwise credible interval of $f$ at $x$ is the interval $I\_x := [ \\mathbb E\_{f\\sim\\pi\_{f|\\mathcal{D}}} f(x) - s, \\mathbb E\_{f\\sim\\pi\_{f|\\mathcal{D}}} f(x) + s]$ such that $\\mathbb{P}\_{f\\sim\\pi\_{f|\\mathcal{D}}} [f(x) \\in I\_x] = 0.8$.
> > > > > > > - As we explain for your next question, we only need to access the ground truth posterior through its score function.
> > > > > > >
> > > > > > > We have revised that subsection to provide better clarity on the above points.
> > > > > > >
> > > > > > > > **[Q21b.]**  I don’t understand what eq 1 means. Does this have something to do with KSD? I general I couldn’t follow the KSD stuff.
> > > > > > > >
> > > > > > > > **[Q21c.]**  Fig2: I don’t know what “LD predictive distribution” or “approximate functoin space posterior” means.
> > > > > > >
> > > > > > > The revision included more discussion about the estimation of the KSD.
> > > > > > >
> > > > > > > - We use KSD to measure the discrepancy between the distribution of $(\\mathcal A\\theta\_t)(\\mathbf X\_e)$ and the groundtruth distribution $p(f(\\mathbf X\_e))$ of $f(\\mathbf X\_e)$ with $f \\sim \\pi\_{f \\mid \\mathcal D}$.
> > > > > > > - The calculation of KSD only requires the specification of a kernel in an Euclidean space, samples from $(\\mathcal A\\theta\_t)(\\mathbf X\_e)$, and the score function of $p(f(\\mathbf X\_e))$.
> > > > > > > - Eq. (1) (which becomes Eq. (3) in revision) decomposes the required score function into two terms. The first term is the score function of the prior, which can be estimated by prior samples; and the second term can be directly evaluated.
> > > > > > > - "LD predictive distribution" refers to the distribution of $(\\mathcal A\\theta\_t)(\\mathbf X\_e)$, and "approximate function space posterior" refers to the distribution $p(f(\\mathbf X\_e))$. We have made these clear in revision.
> > > > > > >
> > > > > > > > **[Q22.]** Earlier H was a group, now its width.
> > > > > > >
> > > > > > > Thanks for pointing this out. We have changed the notation for NN width to $W$.
> > > > > > >
> > > > > > > > **[Q23.]** The paper talks about LD, but experiments are using MALA. Is this ok? [...] MALA is also undefined.
> > > > > > >
> > > > > > > - We cannot directly "implement LD" as you suggested: it is a continuous time process and its simulation has to come with discretization errors. We use Metropolis-Adjusted Langevin Algorithm (MALA) to adjust for this.
> > > > > > > - As the revision now better clarifies (see our response to your Q4 for a summary), our experiment does not directly validate the theoretical results, but rather aims at evaluating the efficacy of practical, Langevin-type algorithms on practical BNN models. MALA is one of these commonly used algorithms. The revision has added the results for ULA (unadjusted Langevin algorithm) in Section 3.2, which is another practical algorithm.
> > > > > > > - The revision has clarified on the above points at the beginning of Section 3 and added references to MALA.
> > > > > > >
> > > > > > > > **[Q24.]** So it seems [from Figure 1] that we get better fits with more samples.  What research question or claim are you studying here?
> > > > > > >
> > > > > > > 1. There isn't any mention of sample size in Figure 1. Fig. 1 visualizes the approximate posterior as the number of MALA iteration, $I$, increases. It suggests that the approximate function-space posteriors converge at a similar rate across two BNN models with a different width (i.e., level of overparametrization).
> > > > > > > 2. As stated in the subsection, Fig. 1 and Fig. 2 suggests that on the practical feed-forward network models, the convergence of the function-space posterior of Langevin-type algorithms may still be unimpeded by overparametrization. At a high level, the demonstration of a result shares the same goal as our theoretical analysis, which is to understand the efficacy of Langevin-type algorithms on BNNs (see our response to your Q4).

---

> > > > > > > > ### Author Response · Authors · 2023-06-19
> > > > > > > > **Response (Part 8)**
> > > > > > > >
> > > > > > > >
> > > > > > > > > **[Q25.]** What is the point of [Figure 2], and what research question are you demonstrating or studying here?
> > > > > > > >
> > > > > > > > 1. Fig.2 plots the KSD against the number of MALA iterates and shows a similar phenomenon as Fig.1: across different models the function-space posterior converges at a similar rate. See our response to Q24 above for the research question.
> > > > > > > > 2. We have two figures for this phenomenon because as stated in the subsection, KSD estimation is more informative (it provides information about multi-dimensional marginals), but is only possible at a smaller scale.
> > > > > > > >
> > > > > > > > > **[Q26a.]** In 3.2. I can’t follow what you do. It seems that you sample [...] a lot of ground truth functions. Yet do you only do MALA once..? Are [the multiple samples for $\\theta_0$] the ground truth functions, or draws from the prior?
> > > > > > > > > **[Q26b.]** It would be very helpful to have algorithm boxes to describe the experiments.
> > > > > > > >
> > > > > > > > - The goal is to estimate the average-case risk, Eq (6) (was Eq. 4 in the original submission), for the estimator derived from MALA approximated posteriors. It (indirectly) reflects sample quality, because a similar estimator derived from the exact posterior would minimize such a risk.
> > > > > > > > - As can be seen from its definition, to construct a Monte Carlo estimate for that risk, we need to draw multiple ground truth functions (equivalently $\\theta\_0$) from the prior; then, *for each sampled $\\theta\_0$*, we use it to generate $y\_i|x\_i$, and run MALA separately for that $\\theta\_0$.
> > > > > > > > - We have added an algorithm box, Algorithm 1 in Appendix C.
> > > > > > > >
> > > > > > > > > **[Q27.]**  I also wonder about the mean arguments: surely in a multimodal posterior you should not average since you can end up between peaks.
> > > > > > > >
> > > > > > > > - The "posterior mean estimator" referred to averaging the prediction function, not the NN parameters; the former is Bayes-optimal for the Gaussian likelihood. We have revised the text to better emphasize this point.
> > > > > > > > - Upon reflection, we realized that for Laplace likelihood, we should have approximated a similarly defined median estimator (see Alg. 1). We apologize for the inconvenience. We have updated the description and experiment results accordingly; the results are extremely close, e.g., in Table 1 and Table 2 the maximum change in MAE was 0.02, and most changes are $\\le 0.01$.
> > > > > > > >
> > > > > > > > > **[Q28.]** I’m having in general trouble seeing the research question studied in 3.2.
> > > > > > > >
> > > > > > > > * The ultimate goal is to understand the efficacy of Langevin-type algorithms on BNNs; see our response to your Q4a.
> > > > > > > > * More specifically, this experiment evaluates the predictive performance *in a setting that eliminates model misspecification*. As discussed in our response to your Q27, in such a setup the predictive performance reflects sample quality.
> > > > > > > > * See our response to your Q4b for the differences from Izmailov et al (2021).

---

> > ### Comment · Reviewer_seUs · 2023-06-20
> > **resp**
> >
> > Thanks for the response.
> >
> > I am still confused of the main claims and contributions. Somehow I’m not getting the main points despite your best efforts, and it’s especially difficult to see how the results relate to other ML research in this area.
> >
> > I’m commenting here your points 1-4 to my Q1a.
> >
> > I see that WLD and FLD is equivalent, which is nice. But I don’t see why this means that WLD does not suffer from overparameterisation. If WLD and FLD are equivalent, and WLD suffers from overparam, then surely FLD should suffer also. Equally, if WLD and FLD are equivalent, and FLD does not suffer from overparam, then surely WLD should not suffer either. These are a contradiction, and I wonder if you can help me understand why the reasoning only goes one way?
> >
> > As an example, let’s take the simple overparameterised model: scalar real input $x$, scalar real output $f(x) = \sum_{i=1}^{1000} w_i x$, with thousand real coefficients ${w_i}$. Here only the sum of $w_i$'s matters, yet the WLD would do tons of work sampling the entire 1000-dim space of weights. I think you argue that in this situation no overparam issues arise in WLD. Really? Can you show that this also is true empirically?
> >
> > You mention that WLD doing function space inference is worth noting. This feels obvious and trivial. Why is this worth noting? Isn’t this a trivial assumption made by all (SG)LD works in this domain? You also mention “apparent benefits”, and “limitations” in existing works. Can you elaborate a bit what these are? You also mention “concrete insights”. Can you give some examples?
> >
> > Finally, two more questions.
> >
> > In example 2.1. You define $\cal{A}$ as $A$, which is matrix of size (K,d). This means that we take the d-dim parameters, and multiply by A to get K-dim outputs. Ok, but when do the inputs $x$ enter the picture? Do we look at the inputs $x$ at all? How do we evaluate $f(x)$? Furthermore, each row of $A$ can be different. So for each input, we multiply the parameters with a different row of $A$. This does not seem to make sense? It would be useful to write down the function form of the $f(x)$ in this example. For instance, is it $f(x) = Ax$ or $f(x) = A \theta x$ or $f(x) = A \theta$? Also, I wonder what is the size of inputs $x$ in this example.
> >
> > Are there any assumptions on $\cal{A}$? For instance, does it need to be smooth (eg. relu is not), or continuous (eg step is not), or deterministic (eg dropout is possibly not)?

---

> > > ### Author Response · Authors · 2023-06-20
> > > **Thanks & Quick Response**
> > >
> > > Thank you for your prompt response. We will carefully consider how to revise the manuscript to best explain these points and we will also include the simulation you suggested, but first we would like to provide some quick repsonse that should help to resolve the confusion.
> > >
> > > > [**1.**] I see that WLD and FLD is equivalent [...] But I don't see why this means that WLD does not suffer from overparameterisation
> > >
> > > Our point is similar to your second case: "if WLD and FLD are equivalent, and FLD does not suffer from overparam, then surely WLD should not suffer". But there is no contradiction: WLD will not suffer *only if we are only interested in its implied function-space posterior*; in our notations, the convergence of the distribution of $\\mathcal{A}\\theta\_t$ to $\\pi\_{f|\\mathcal{D}}=\\mathcal{A}\_\\# \\pi\_{\\theta|\\mathcal{D}}$ can be very fast, but the convergence of $\\theta\_t$ to $\\pi\_{\\theta|\\mathcal{D}}$ is generally still slow. The latter is what "WLD suffers from overparam" means.
> > >
> > > > [**2.**] As an example, let's take the simple overparameterised model
> > >
> > > - This is a nice example and is covered by our Example 2.1 (ii). From that example (which is mathematically proved) we can deduce that, the convergence of the function-space distribution implied by WLD is as fast as the convergence of FLD, which  in this case reduces to a Riemannian LD defined on a one-dimensional space.
> > > - We agree a simulation of this case can be helpful and are working on one to be included in the revision. Meanwhile, the fact that *(function-space inference using) WLD is equivalent to a one-dimensional sampling problem* here should also provide some insights. The proof of the example provides additional intuition: here WLD can be "decomposed" into two independent dynamics, the one-dimensional one that corresponds to the sampling of predictive functions, and a 999-dimensional one which is irrelevant for this purpose, but relevant if we care about the entirity of $\\pi\_{\\theta|\\mathcal{D}}$.
> > > - Your example corresponds to our Ex 2.1 because we can set $(\\mathcal{A}\\theta)(x) = f(x;\\theta) := \\sum\_{i=1}^{1000} \\theta\_i x$ and $\\mathcal{X}=\\{1\\}$. Note that it suffices to consider $\\mathcal{X}=\\{1\\}$ here because for any $\\theta$ and $x\\ne 1$, $f(x;\\theta)$ is determined by $f(1;\\theta)$.
> > >
> > >
> > > > [**3.**] When do the inputs enter the picture? the function form of $f(x)$ in [Example 2.1] what is the size of inputs $x$
> > >
> > > * $\\mathcal{A}$ maps a parameter $\\theta$ to an entire function, $\\mathcal{A}\\theta = f(\\cdot;\\theta) \\in \\mathcal{F}$; the input then enters the picture when we evaluate $(\\mathcal{A}\\theta)(x) = f(x;\\theta)$.
> > > * the function form is generally $f(x;\\theta) = \\theta^\\top \\sigma(x)$, where $\\sigma(x)$ is a predetermined, possibly nonlinear function that maps $x\\in\\mathcal{X}$ to the "features" $\\sigma(x)\\in \\mathbb{R}^d$. Thus, $x$ can have any dimensionality.
> > >
> > > > [**4.**] Are there any assumptions on $\\mathcal{A}$? For instance, does it needs to be smooth (eg. relu is not)
> > >
> > > * For our function-space posterior to be well-defined, $\\mathcal{A}$ only needs to be a measurable (deterministic) map from $\\mathbb{R}^d$ to $\\mathcal{F}$ (w.r.t. any $\\sigma$-algebra on $\\mathcal{F}$). We always use the Borel $\\sigma$-algebra, which only requires the specification of a topology on $\\mathcal{F}$. All our examples can be viewed as using the quotient topology $R^d/H$ (cf. Sec 2.3 for the definition $H$; note our Example 2.1 also has a group structure, which is a linear group).
> > > * As clarified above, $\\mathcal{A}$ is the map from parameters to functions, not the input-output map. So for example, applying ReLU to the inputs will not affect the smoothness of $\\mathcal{A}$.

---

> > > > ### Author Response · Authors · 2023-06-22
> > > > **result for the suggested experiment**
> > > >
> > > > As you suggested, we conducted the following experiment for the model $f(x;\theta) = \sum_{i=1}^d \theta_i x$: we generate 100 training samples, where the input $x_i$ follows the distribution Section 3.1, $y_i|x_i,\theta_0 \sim \mathcal N(f(x_i;\theta_0), 0.5)$, and $\theta_0\sim\pi_\theta$ is sampled from the prior. We use $\pi_\theta := \mathcal N(0, (1/d) I)$ to match the standard scaling in NN models. We simulate 40000 parallel MALA chains to approximate samples from (WLD), and compare the MALA-approximated marginal posterior for $\sum_{i=1}^d \theta_i$ with the respective ground truth distribution, which is available in closed form for this example.  Similar to Fig.2, we report $\sqrt{\text{KSD}}$ for different choices of $d$ and number of MALA iterations. The result is shown in the following table:
> > > >
> > > > | # iterations | 75      | 375   | 1575  | 6375  |
> > > > | ------------ | ------- | ----- | ----- | ----- |
> > > > | $d = 1$      | 151.653 | 2.890 | 0.117 | 0.146 |
> > > > | $d = 10$     | 150.873 | 2.812 | 0.231 | 0.102 |
> > > > | $d = 100$    | 151.284 | 2.644 | 0.236 | 0.157 |
> > > > | $d = 1000$   | 150.509 | 2.882 | 0.160 | 0.079 |
> > > >
> > > > We can see that the convergence speed for $d=10^3$ is similar to that for $d=1$, which is consistent with our analysis (note the KSD estimates still have a random fluctuation).  We have verified visually that at the end of the experiments, the 1D approximate posterior is extremely close to the ground truth.

---

### Review · Reviewer_rKhH · 2023-06-05

**Summary Of Contributions:**

This work illustrates an equivalence between weight-space and function-space Langevin dynamics applied to Bayesian models. The authors explain the general intuition that we can expect weight-space LD to induce LD in function space. They theoretically derive this equivalence in a couple of concrete cases - the linear case and a simplified BNN model - and explain how the intuition from these cases can be expected to generalize to more complex settings. The authors then provide a pair of experiments, one on a 1D synthetic dataset which aims to validate the prediction made by this equivalence that the rate of convergence to the ground truth should not depend on the degree of overparameterization, and the second on a semi-synthetic dataset which aims to demonstrate that weight-space LD approximates the function space posterior well.

Overall, I believe that the theoretical contributions of the work are interesting and novel as far as I am aware. I agree with the sentiment implied by the authors that the primary value of the work is not in its theoretical complexity, but rather in the high-level intuition that it provides for understanding and interpreting Langevin dynamics methods and theoretical substantiation of that intuition. I find the experimental claims of the paper less compelling - the experiments provide some support to the authors' claims that weight-space LD correctly approaches the functional posterior and doesn't depend on the degree of overparameterization, but I don't think this is sufficient to convincingly demonstrate the broader equivalence between WLD and FLD in practice.

**Audience:**

Yes

**Broader Impact Concerns:**

I have no broader impact concerns.

**Claims And Evidence:**

Yes

**Requested Changes:**

The principle change I would like to see is a clarification of what the authors' believe their experiments in Section 3 demonstrate. In particular, if they plan to claim it then it would be beneficial to include an experiment that showed the dynamics of WLD and FLD (rather than just their endpoints) were equivalent in practice.

The rest of my suggested changes are minor, but would strengthen the work in my view:
- In Example 2.1(i), the authors say that the derivation is given in Appendix 2.1. However, this appendix only seems to include the proof of Proposition 2.1. Is this proof of Proposition 2.1 supposed to also indicate what happens in Example 2.1? If so, I'm not sure I understand the connection. If not, then it might be worth indicating where the results in Example 2.1(i) come from.
- I would consider providing more details of Example 2.1(ii), perhaps in an appendix, since the current explanation is quite brief, and this is the only place where continuous symmetries of the parameter space are explicitly dealt with. For instance, it might be worth making explicit that (as I understand it) the fact that the evolution factorizes into two subspaces is dependent on the Gaussian choice of prior.
- The choice of notation $T_p(H \cdot p)^\perp$ is a little confusing to me. Firstly, $H \cdot p$ is a set, and I'm not sure what is meant by the tangent plane at a set (as opposed to a particular point). It might also be worth being explicit about the definition of the symbols $^\perp$ and $\mathrm{d} \mathcal{A}|_p u$.
- I'm not sure what is meant by "let $\pi_\theta$ have constant density'' in Example 2.2, since it's defined on $\mathbb{R}^d$ with respect to the Lebesgue measure - where the constant measure isn't a probability measure.
- In Example 2.3, the section "The evolution of $\tilde p_t$ ... of equivalence classes." is quite vague. It's not clear what the phrases "closely related to'' refers to, or exactly what fast vs. slow convergence entails in this context. It would be nice to make more precise what is meant here.
- The authors could consider clarifying the argument in Section 3.2 for why this experiment provides evidence about the equivalence of WLD and FLD. (As mentioned above, this was a point of confusion for me.)
- In Appendix A.1 there is a clash of notation where $g$ is used to denote both the metric tensor and a test function.
- It might be worth defining the notation $\mathrm{blkdiag}(P_1, \dots, P_{m_x})$.

**Strengths And Weaknesses:**

### Strengths

- The authors provide lots of intuition for how to interpret their theoretical results, which is much appreciated given that this is in my view the main contribution of the paper. The authors also make good reference to the previous literature and explain well how their results develop previous work.
- The work is very clear throughout and easy to follow. The authors' choice of illustrative examples in Section 2 are well chosen and significantly aid the communication of their overarching thesis. The detailed derivation of Example 2.3 is also enlightening.

### Weaknesses

- The gap between the assumptions made in Proposition 2.1 and the conditions that we expect to hold for real neural networks is quite large. For example, most neural networks will have at least some continuous symmetries. Although the authors' mention continuous symmetries in Remark 2.1, it is still quite unclear to me how large the gap between reality and the assumptions made in this paper is. The authors might consider further indicating either why they think their assumptions are nevertheless reasonable, or how large they consider the gap to be.
- I'm not convinced that the experiments given in Section 3 convincingly demonstrate the equivalence between WLD and FLD that the authors claim they do and which is explained from a theoretical perspective in Section 2. I think the authors equivocate in different places between claiming that (a) weight space LD approaches the functional posterior at a rate that's independent of overparameterization and (b) WLD is equivalent to FLD. My interpretation of the results are that Section 3.1 shows that (in this synthetic case) WLD seems to converge in function space at a rate that's independent of the degree of overparameterisation, and that Section 3.2 shows that the FLD-approximated posterior is close to the true posterior.
- The experiment performed in Section 3.1 is extremely small (with only 3 datapoints in one version, as I understand it). I'm not sure this is big enough for us to learn anything meaningful, especially with regards to deep neural networks as they are used in practice.
- The conclusions the authors draw from the experiment in Section 3.2 seem to me to go via an argument that gradient descent performs well on this task and therefore must closely approximate the exact posterior mean in function space, so if MALA and GD are getting similar performance then this means that the MALA-approximated posterior must also be cloes to the function-space posterior. This argument seems a bit tenuous to me, and relying on assumptions about behaviors of GD that are unsubstantiated.

---

> ### Author Response · Authors · 2023-06-19
> **Response (Part 1)**
>
> Thank you for your thoughtful review; we are glad you appreciated the contributions of this work. Below we address your questions in detail.
>
> ### Main Concerns
>
> > **[Q1.]** The gap between the assumptions made in Proposition 2.1 and the conditions that we expect to hold for real neural networks is quite large. [...] The authors might consider further indicating either why they think their assumptions are nevertheless reasonable, or how large they consider the gap to be.
>
> * We agree there is a non-trivial gap. We mentioned that we are only treating simplified models, and have now updated the newly formed Section 2.2 and the paragraph above Example 2.3 to better emphasize this point.
> * Note that Proposition 2.1 does not cover NN models; in that subsection Example 2.3 fulfills this purpose. And while that example does not allow for continuous symmetries, Example 2.1, which covers random feature models and thus (formally) NNs in the kernel regime, allows for continuous symmetries. We have updated Section 1, the beginning of Section 2 and Section 2.2 to provide better visibility to that example.
>
> > **[Q2a.]** I'm not convinced that the experiments given in Section 3 convincingly demonstrate the equivalence between WLD and FLD that the authors claim they do.
>
> We apologize for the confusion and have removed all sentences that could create such an impression. Your understanding of the experiments are correct. We will explain their goals in our response to your Q2d.
>
> > **[Q2b.]** The experiment performed in Section 3.1 is extremely small (with only 3 datapoints in one version, as I understand it). I'm not sure this is big enough for us to learn anything meaningful.
>
> * We can only estimate the convergence of function-space marginals at such a scale, due to the challenges in score estimation which needs to generalize out-of-sample. We have updated Section 3.1 to better emphasize this point.
> * Nonetheless, please note that Section 3.1 still uses practical NN models (for tabular data) with up to $10^4$ parameters and complex symmetric structures. The convergence of marginal function-space posteriors, however small their dimension is, cannot be explained by the possible convergence of the weight-space posterior. In this aspect the results provide support to our general message.
>
> > **[Q2c.]** The conclusions the authors draw from the experiment in Section 3.2 seem to me to go via an argument that gradient descent performs well on this task [...]. This argument seems a bit tenuous to me, and relying on assumptions about behaviors of GD that are unsubstantiated.
>
> * We have added some references for past works that analyzed the convergence of GD on overparameterized NN models. While they cannot cover practical NN models, they provide theoretical evidence that GD can be unaffected by overparameterization, which is also consistent with the ample empirical evidence in deep learning applications.
> * To provide further evidence on the efficacy of GD (and LD), we have also added plots for the Bayes errors (which is only attainable given infinite samples) in the experiments. We can see that both GD and the Langevin-type algorithm achieves an error that is close to the Bayes error, especially in problems with a larger sample size.
>
> > **[Q2d.]** The principle change I would like to see is a clarification of what the authors' believe their experiments in Section 3 demonstrate.
>
> We have updated Section 1, Section 3 (the beginning and Sec. 3.2) and Section 4 to clarify on this:
> * The experiments aim at evaluating the convergence of the function-space distributions of the weight-space Langevin-type algorithms. They do not provide direct evidence for the equivalence, but rather for a *possible consequence* of it: when the model is correctly specified and the metric in (FLD) is not known to have an adverse effect, the function-space distributions may have good convergence properties (see Sec 2.2).
> * These experiments also share a high-level goal with our analysis, which is to understand the behavior of Langevin-type algorithms on BNNs. In this aspect, they also complement past works with a similar goal, by analyzing practical models (instead of infinitely wide models as in past theoretical works) and by eliminating the influence of model misspecification (which appears in the past empirical works cited in paper).

---

> > ### Author Response · Authors · 2023-06-19
> > **Response (Part 2)**
> >
> >
> > ### Suggestions of Minor Changes
> >
> > Thank you for your suggestions; we will respond to them in turn.
> >
> > >  **[Q3a.]** Is the proof of Proposition 2.1 supposed to also indicate what happens in Example 2.1?
> >
> > Yes. We have added in Appendix B.1 that Example 2.1 (i) can be proved by invoking Proposition 2.1 (b).
> >
> > >  **[Q3b.]** I would consider providing more details of Example 2.1(ii), perhaps in an appendix.
> >
> > We added the full proof of Example 2.1(ii) in Appendix B.1. As you understand, the Gaussian prior is important in the factorization because its gradient is linear; see the last equation in the proof.
> >
> > > **[Q4.]** It might also be worth being explicit about the definition of the symbols $\\perp$ and $\\mathrm{d} \\mathcal A|\_p u$.
> >
> > We have added the definitions below the display formula where these symbols first appeared. In more details:
> >
> > 1) The orbit $H\\cdot p$ is a submanifold of $\\mathbb R^d$. Its tangent space at $p$ is $T\_p(H \\cdot p)$ and can be viewed as a subspace of $T\_p \\mathbb R^d$. Given the Riemannian metric, the orthogonal complement $T\_p(H \\cdot p)^\\perp$ can be defined as $\\{ v \\in T\_p \\mathbb R^d : \\langle v, w \\rangle = 0, \\forall w \\in T\_p(H \\cdot p) \\}$.
> > 2) $\\mathrm d \\mathcal A|\_p$ is the differential of $\\mathcal A$ at $p$. It is a linear map $T\_p \\mathbb R^d \\to T\_{\\mathcal Ap} \\mathcal F$. Therefore, $\\mathrm d \\mathcal A|\_p u$ is a tangent vector in the tangent space of $\\mathcal F$ at $\\mathcal Ap$. We changed this notation to $(\\mathrm d \\mathcal A|\_p)(u)$ for clarity.
> >
> > > **[Q5.]** I'm not sure what is meant by "let $\\pi\_\\theta$ have constant density'' in Example 2.2, since [...] the constant measure isn't a probability measure.
> >
> > Here we are discussing an improper prior. We have updated Proposition 2.1 to note that the prior may be improper.

---

> > > ### Author Response · Authors · 2023-06-19
> > > **Response (Part 3)**
> > >
> > >
> > > > **[Q6.]** In Example 2.3, [...] it's not clear what the phrases "closely related to'' refers to, or exactly what fast vs. slow convergence entails in this context.
> > >
> > > We have added a footnote to clarify that
> > > 1.  Eq (2) in the example is "closely related" to reflected LD in the following sense: we can easily prove that Eq (2) describes the density evolution of the reflected LD if the domain is bounded with smooth boundary (see the following). In our case, however, the boundary does not satisfy such properties. Proving their correspondence in our case takes more effort, yet the benefit is somewhat unclear. Thus, we have chosen such a phrase.
> > > 2. Reflected LD may have "fast convergence" in that setting: a proof for convex and bounded domain with smooth boundary can be found in Bubeck et al. (2018). We noted that weight-space LD will have to explore an exponential number of modes in that setting; thus, its convergence can be much slower than that of $\\tilde p\_t$ (in function space). Indeed, when the modes are well-separated, the exploration can take an exponentially long time (Lee et al, 2018; Bovier et al. 2004).
> > >
> > > We now present the proof for bounded and smooth boundary.
> > >
> > > Suppose $X\_t$ follows the reflected Langevin dynamics:
> > >
> > > $$ d X\_t = -\\nabla V(X\_t) d t + \\sqrt 2 d B\_t + \\vec n(X\_t) L(dt),$$
> > >
> > > where $V: \\Omega \\to \\mathbb R$ is an energy function (e.g., $\\log p(\\mathbf Y | \\theta, \\mathbf X) + \\log p\_\\theta(\\theta)$ in our case), $\\Omega$ is a bounded domain with smooth boundary, $\\vec n(X\_t)$ is the outward normal vector at $X\_t$, and $L$ is the local time (see Sato et al. (2022)).
> > >
> > > The generator $\\mathcal L$ of the above dynamics is
> > >
> > > $$\\mathcal L f = -(\\nabla V)^\\top \\nabla f + \\Delta f,$$
> > >
> > > where $f: \\Omega \\to \\mathbb R$ satisfies the Neumann boundary condition $\\vec n^\\top\\nabla f = 0 \\text{ on } \\partial \\Omega$ (see Moitra and Risteski (2020, p.15)). Let $q\_t$ be the distribution of $X\_t$, then it follows the Kolmogorov forward equation $\\partial\_t q\_t = \\mathcal L^\* q\_t$, where $\\mathcal L^\*$ is the adjoint operator such that $\\int\_\\Omega \\mathcal Lf g = \\int\_\\Omega f \\mathcal L^\*g$ holds for every $f \\in \\mathcal D(\\mathcal L), g \\in \\mathcal D(\\mathcal L^\*)$, and $\\mathcal D(\\mathcal L)$, $\\mathcal D(\\mathcal L^\*)$ represent the domains of the generator and its adjoint, respectively (see Pavliotis (2014, Chap. 2)).
> > >
> > > By the divergence theorem (Maggi (2012, Remark 9.5)), we know
> > >
> > > $$ \\int\_\\Omega \\mathcal Lf g = \\int\_\\Omega f \\left (  \\nabla \\cdot ( g \\nabla V) + \\Delta g  \\right ) + \\int\_{\\partial \\Omega} \\vec n^\\top \\left ( g \\nabla f - f \\nabla g - fg \\nabla V  \\right ).$$
> > >
> > > In our example, $V$ is invariant under group action so $V(x) = \\frac{1}{|S\_n|}\\sum\_{\\varphi \\in S\_n} V(\\varphi \\cdot x)$. Lemma B.1 then implies $\\vec n^\\top \\nabla V = 0$. Now, it is clear that to make the condition  $\\int\_\\Omega \\mathcal Lf g = \\int\_\\Omega f \\mathcal L^\*g$ holds (note $\\vec n^\\top f = 0$ for $f \\in \\mathcal D(\\mathcal L)$), $g \\in \\mathcal D(\\mathcal L^\*)$ should satisfy the Neumann boundary condition $\\vec n^\\top \\nabla g = 0$ and $\\mathcal L^\* g = \\nabla \\cdot (g \\nabla V) + \\Delta g$. This shows that $q\_t$ also satisfies the equation for $\\tilde p\_t$ in Example 2.3.
> > >
> > > > **[Q7.]** The authors could consider clarifying the argument in Section 3.2 for why this experiment provides evidence about the equivalence of WLD and FLD.
> > >
> > > We have removed such claims and updated the explanation about the goal of the experiments (see our response to your Q2).
> > >
> > > > **[Q8.]** In Appendix A.1 there is a clash of notation where $g$ is used to denote both the metric tensor and a test function.
> > >
> > > We have changed the notation of test functions from $g$ to $\\zeta$.
> > >
> > > > **[Q9]** It might be worth defining the notation $\\mathrm{blkdiag}(P\_1, .., P\_{m\_x})$.
> > >
> > > We have added a footnote: it refers to a block diagonal matrix with diagonal elements $P\_1, ..., P\_{m\_x}$.
> > >
> > > ### References
> > >
> > > [1] Pavliotis, Grigorios A. Stochastic processes and applications: diffusion processes, the Fokker-Planck and Langevin equations. Vol. 60. Springer, 2014.
> > >
> > > [2] Maggi, Francesco. Sets of finite perimeter and geometric variational problems: an introduction to Geometric Measure Theory. No. 135. Cambridge University Press, 2012.
> > >
> > > [3] Bovier, Anton, et al. "Metastability in reversible diffusion processes I: Sharp asymptotics for capacities and exit times." Journal of the European Mathematical Society 6.4 (2004): 399-424.
> > >
> > > [4] Lee, Holden, Andrej Risteski, and Rong Ge. "Beyond log-concavity: Provable guarantees for sampling multi-modal distributions using simulated tempering langevin monte carlo." Advances in neural information processing systems 31 (2018).

---

> > > > ### Comment · Reviewer_rKhH · 2023-06-26
> > > > **Thanks for your thorough response**
> > > >
> > > > Thank you very much for your thorough response. I particularly appreciate the clarification in the manuscript with regards to what the authors claim that their experiments demonstrate.
> > > >
> > > > I believe that the submission reads quite clearly now, and the main remaining concerns seem to be around the extent to which the author's claims are interersting and meaningful versus trivial or already known. Personally, I'm inclined to see the theoretical results as a somewhat interesting incremental addition to the liteature.
> > > >
> > > > For the experimental work, while I appreciate the clarification that the experiments do not aim to directly show the equivalence of WLD and FLD, it is a bit unfortunate that the authors have no way of experimentally testing this claim, since it seems to me a key motivating point of their paper - though I understand their point that testing this claim is difficult as they cannot perform function-space inference directly. As it stands, the experiments still seem to me to be slightly detatched from the core story the authors want to tell, and I'm not convinced that they constitute a meaningful contribution over the existing work.

---

### Review · Reviewer_HLTU · 2023-06-08

**Summary Of Contributions:**

The reviewed paper is concerned with inferential problems where (a) the model
is over-parametrised (e.g. Bayesian neural networks); and (b) Langevin dynamics
are used to sample from the corresponding posterior. It discusses in
particular the situation where the Langevin algorithm seems to mix poorly in
the original parameter space, yet it may mix sufficiently well in a
lower-dimensional space, which may be sufficient for approximating the
posterior predictive distribution.
The main point of the paper seems to derive mathematical results to explain
this phenomenon.




**Audience:**

No

**Broader Impact Concerns:**

No broader concern.

**Claims And Evidence:**

Yes

**Requested Changes:**


Relevant references
===============

It might make to cite some of the papers belows, and others cited therein.

The following papers discuss the interplay between (over-)parametrisation and
MCMC mixing in various ways:

* Papaspiliopoulos, Roberts, and Sköld (2007). A general framework for the
  parametrization of hierarchical models. Statist. Sci. 22, no. 1, 59–73.

* Yu and Meng (2011).  To center or not to center: that is not the question—an
  ancillarity-sufficiency interweaving strategy (ASIS) for boosting MCMC
  efficiency. J. Comput. Graph. Statist. 20, no. 3, 531–570.

Mixture models are a class of model where group actions and invariance plays an
important role:

* Frühwirth-Schnatter (2001). Markov chain Monte Carlo estimation of classical
  and dynamic switching and mixture models.  J. Amer. Statist. Assoc. 96, no.
  453, 194–209.

* Celeux, Hurn and, Robert (2000). Computational and inferential difficulties
  with mixture posterior distributions.  J. Amer. Statist. Assoc. 95, no. 451,
  957–970.

**Strengths And Weaknesses:**

The paper is somehow interesting, but it has the following limitations. First,
the main scientific message (it's ok if your Langevin sampler does not visit
the whole support of the parameter space, as it may have already visited a
portion of the support which is sufficiently representative, as far as the
predictive distribution is concerned) has been stated before in various recent
papers (cited on the first page). Second, the mathematical results derived in
the paper to support this message actually say much less that they authors want
them to say:

* A minor point, but a function space is usually infinite-dimensional, and then
  it becomes non-trivial to define a probability distribution with respect to
  such a space, not to mention defining Langevin dynamics. But, in this paper,
  the "function space" is actually of dimension $k\leq d$, the dimension of the
  parameter space, see bottom of p. 2. It would be more sensible to say that we
  are considering over-parametrised models rather than "function spaces".
  (Incidently, there are quite a few papers that discusses already MCMC and
  related algorithms for over-parametrised models, or more generally how
  parametrisation may affect MCMC, but they are not cited; see section below).

* The mathematical results say essentially: if I generate a Langevin diffusion
  associated to the posterior of $\theta$ (the parameter in the original
  parametrisation), then, in certain special cases, the trajectory of $f$ (the
  parameter in the transformed space) is also a certain Langevin diffusion.
  Fine, but this is almost orthogonal to the point of interest: it's not
  because a trajectory follows Langevin dynamics that it necessarily mixes
  well, and vice-versa.

* If I know already that the model is over-parametrised, then what would make
  more sense practically would be to use some pre-conditioner for the WLD
  diffusion. Say my model is such that the parameter is $\theta=(\theta_1,
  \theta_2)$ and, the likelihood depends only on the sum $\theta_1+\theta_2$.
  Then the corresponding posterior will be very elongated along the line
  $y=-x$, but, if I rotate the axes (i.e consider parameters $\eta_1 =
  \theta_1+\theta_2$ and $\eta_2 = \theta_1 - \theta_2$), then I may get much
  better mixing, both for $\theta$, and for the parameter of interest, which is
  $f=\eta_1$ here.

* The first part of the results concerns the case where the transformation
  (from $\theta$ to $f$) is actually linear. Sorry, but this case is completely
  trivial.

* The second part of the results concerns group actions, and is more
  interesting, see in particular the torus in Example 2.2. Still, it looks like

  (a) the actually results are just pulled from other references (Villani,
  2009); and

  (b) they may not be so relevant in practice, as, Bayesian
  neural networks and similar over-parametrised models do not seem to have this
  particular group structure in most cases.

---

> ### Comment · Reviewer_HLTU · 2023-06-09
> **Missing word**
>
> In "Relevant references" :
> It might make to cite => It might make sense to cite
>
> Sorry.

---

> ### Author Response · Authors · 2023-06-19
> **Response (Part 1)**
>
> Thank you for your thoughtful review. We will address your questions in turn. We will also clarify on some important misunderstandings around the contribution of our work, and have revised our submission to provide greater clarity on these points.
>
> ## On our contributions and the related questions
>
> First we would like to clarify on **our contributions**.
> * The main takeaway from our work is *not* that Langevin dynamics (LD) mixes well in (some) overparameterized models; such a result has been established in prior work, as cited in Section 1 and noted by you. Our goal is to show that in various overparameterized models, weight-space Langevin dynamics (WLD) is equivalent to a function-space LD or its variants ("FLD" henceforward) and can thus be viewed as a "function-space inference" procedure.
> * We will explain the relevance of such equivalence results in detail in our following answer. Here we also note that they are most interesting in the context of BNN models, which have motivated the study of such function-space procedures, and two models we analyzed (Ex. 2.1 and Ex. 2.3) are chosen to be relevant for this purpose; see our response to Q3 and Q4b below.
>
> We now address your related questions.
>
> > [**Q1.**] It is not because a trajectory follows LD that it necessarily mixes well.
>
> - We agree, and our focus is on the former. We have removed a paragraph below Remark 2.1 which may have created the misunderstanding.
> - The main reason we focus on the former point is that such a result still shows that the convergence of (the function-space pushforward of) WLD is *not directly influenced by overparameterization*: it is equivalent to FLD which does not have to defined w.r.t. an overparameterized model, even though the mixing of FLD is not necessarily desirable. As discussed in Section 1, this property has been the design goal of the various "function-space inference" procedures in the past, and *the fact that a simple and commonly used algorithm such as WLD already fulfills this property is worth noting*.
> - Such a result also provides *intuitive understanding for the behavior of WLD*. As we noted in the submission, its equivalent FLD is defined with a pushforward of the weight-space Euclidean metric. For NN models, this metric is known as the neural tangent kernel (NTK) and has been extensively studied. The preconditioning effect of NTK is often considered desirable, but there are also models for which we can know *a priori* that the NTK preconditioning may be harmful, such as very deep NNs in a certain initialization regime.  *In either case, our intuition about FLD can be transferred to WLD if the equivalence result holds.*  We have reorganized the relevant discussions into a new subsection, in Section 2.2.
>
> > [**Q2.**] If I know already that the model is over-parametrised, then what would make more sense practically would be to use some pre-conditioner
>
> * This will not solve all problems stemming from overparameterization: preconditioning cannot address the discrete symmetries in the model, which is particularly relevant for BNNs.
> * As discussed in our answer to your Q1, our WLD corresponds to an FLD which is (implicitly) preconditioned by the NTK, and for many NN models such a preconditioning effect is considered desirable.
>
> > [**Q3.**] The first part of the results concerns the case where the transformation (from $\\theta$ to $f$) is actually linear. Sorry, but this case is completely trivial.
>
> * The relevance of this example is not derived from its technical complexity; *it is from its connection to random feature models, and thus (formally) neural networks in the "kernel regime"*, as we noted below the example.  As discussed in the newly formed Section 2.2 (and summarized in our response to your Q1), the connection provides insights into the behavior of WLD.
> * This example is also relevant because some popular inference schemes for BNNs are only justified in this regime; see the penultimate paragraph of Section 1. We have added another sentence below the example, to better highlight this point.

---

> > ### Author Response · Authors · 2023-06-19
> > **Response (Part 2)**
> >
> >
> > > [**Q4a.**] the actually results [concerning group actions] are just pulled from other references (Villani, 2009)
> >
> > The goal of our analyses is to establish the equivalence between WLD and FLD; these equivalence results are proved by ourselves. We have referenced past works such as Villani (2009) only to show that FLD can often have good mixing guarantees.
> >
> > > [**Q4b.**] they may not be so relevant in practice, as, Bayesian neural networks and similar over-parametrised models do not seem to have this particular group structure in most cases
> >
> > * We agree that practical BNN models are much more complex and not covered by our analyses. For this reason we studied their behavior through indirect numerical experiments. We have revised the beginning of Section 3 and Section 4 to better emphasize this point.
> > * Past analyses for DNNs often focus on similar simplified models. Our setup is inspired by Wei et al (2019); despite its simplicity it is still a step further from the commonly studied infinitely-wide NN models in the "kernel regime" (Jacot et al, 2018), as it allows for a varying NTK.
> >
> > ## Response to the other questions
> >
> > > [**Q5.**] It would be more sensible to say that we are considering over-parametrised models rather than "function spaces".
> >
> > * We use the term "function space" to be consistent with the recent works on "function-space inference" using overparameterized NN models, and to emphasize the difference between model parameters (which exhibit symmetries) and functions.
> > * Throughout our paper, the model defines a valid finite-dimensional space of prediction functions, a prior $\\pi\_f$ on that space, and consequently a valid posterior. The issues with infinite-dimensional priors do not appear in our work. Although related infinite-dimensional formulations are sometimes related to BNN models which is the focus of our work (e.g. ultrawide BNNs converges to GPs), such a discussion appears unnecessary for our goal of understanding the behavior of WLD. Therefore, we forgo it for brevity.
> >
> > > [**Q6.**] It might make to cite some of the papers belows, and others cited therein.
> >
> > Thank you for the references. As now discussed in Section 1 and Section 2.3 in the revision, our work is related to them at a high level but have a different goal:
> > * past works on mixture models, such as the two you have referenced, typically focus on improve mixing in the parameter (i.e., weight) space, whereas our equivalence result only possibly has implications for the function-space marginal distribution.
> > * Similar to our work, Papaspiliopoulos et al (2007) and Yu and Meng (2011) also studied the convergence of a marginal distribution when the model parameterization demonstrates some redundancy. However, their focus is on designing new inference schemes, whereas our work revisits an existing inference scheme which has been extensively used in practice.

---

> > > ### Comment · Reviewer_HLTU · 2023-06-20
> > >
> > > Thanks for your reply, however I am still unconvinced.
> > > While replying to another referee, you said: "The ultimate goal is to understand the efficacy of Langevin-type algorithms on BNNs".
> > > However, I don't see anything in this paper that achieves this goal. Again, it's not because the simulated trajectory follows Langevin dynamics (both in the original space and in the functional space) that it says anything about "the efficacy of Langevin algorithms on BNNs".

---

> > > > ### Author Response · Authors · 2023-06-20
> > > > **Thanks & Quick Response**
> > > >
> > > > Thank you for your prompt response. Please note that
> > > > * the equivalence does allow us to understand the efficacy of WLD *in some cases*, such as when the NTK of the BNN model agrees with the limiting NNGP kernel. This is explained in Section 2.2 in the revision and summarized in our response.
> > > > * the equivalence always allow us to *partially understand the efficacy*, as it shows that WLD is not hindered *by the mere fact* that it is applied to an overparameterized model. This is relevant because as we explained, some of the past works on BNNs have been motivated by a similar understanding that overparameterization generally poses challenges to approximate inference.
> > > >
> > > > Therefore, we believe it is reasonable to say that both our analysis and experiments share this *high-level* goal, although we are happy to consider further alternations to the wording.  Please also note that we have explained our main contribution in the context of approximate inference for BNNs, which appeared to have been a main concern.

---

### Author Response · Authors · 2023-06-19
**Summary of the key changes.**

We thank all reviewers for their detailed and thoughtful feedback.  We have revised the manuscript thoroughly incorporate the feedback from all reviewers, and have uploaded a lightly edited diff file as the supplementary material to highlight the changes.  We also summarize the key changes in the following:

- We have added a top-level paragraph before Section 2.1 to summarize the theoretical discussions and their implications.
- We have noted (Sec.1, Sec.4) that our equivalence results may provide concrete insights into the behavior of weight-space LD on BNNs, by investigating the push-forward metric in the equivalent function-space LD.  We have regrouped the relevant discussions into a new Section 2.2, on which we have also slightly expanded.
- We have clarified that the experiments do not directly validate the equivalence results, but only a possible consequence of it (Section 1, 3 and 4). We have also revised Section 3.1 and 3.2 to clarify on the rationales of our experiments.
- We have clarified on the notations and mathematical notions used throughout the paper, and added a mathematical background section in Appendix A.

---

### Decision · Action_Editors · 2023-07-06

**Recommendation:** Reject

**Comment:**

The reviewers were divided in their recommendation. More important than their recommendation, however, is that they largely agree that the paper is unclear about it claims and aims.

First, the paper needed to be more precise about the nature of the equivalence between weight and function-space Langevin dynamics that the paper is studying, how it is related or not related to efficiency, and what the equivalence allows us to do.

Second, the paper needed to better explain why they can work with a function space that has dimensionality k that is smaller than the number of parameters. The examples in Section 2.1 may be important building blocks for the message of the paper, but currently they are actually more confusing than helpful because the function is assumed to only take on finitely many values which is far removed from actual usage and the framing of the problem in the introduction.

Third, the experimental evidence needed to be better and more clearly support the main message of the paper.


**Audience:**

 The topic of the paper is of interest of the audience of TMLR. However, the current write-up would make it very hard for the interested TMLR reader to understand the paper. This is both because of the muddled message (see above) but also because of the dense mathematical write-up and the insufficient discussion of the results (empirical and theoretical). Reviewers who work on the topic of the paper declined to review since they couldn't well comprehend the work. This and the comments in the reviews indicate that further work is needed to make the paper appropriate for TMLR.



**Claims And Evidence:**

  All reviewers flagged in their initial report that the submitted paper was unclear in its main claims and what the experiments aimed to demonstrate (in particular those in Section 3). The paper has partly improved with the revision, but significant issues do remain and I recommend that the paper undergoes a major revision to clean up its message and to align the empirical evidence with the main claims.

With the major revision, I would ask the authors to take the reviewers' comments fully at heart. The paper now has approximately 10 content pages. I recommend that the authors use the extra space to better explain their results (some figures are barely discussed in the main text) and include more background material on the necessary advanced mathematics (e.g. on Riemmanian manifolds) to make the text more accessible. Moreover, the text needs significant polishing to make the readers' life easier (e.g. all acronyms needed to be defined, notation should be consistent---H vs W for number of layers---etc)


**Resubmission Of Major Revision:**

The authors may consider submitting a major revision at a later time.